# GFD-EMVC: Evolutionary Multi-View Classification with Label Noise via Gradient and Feature Dual-Perception

Shuai Li [1]  Xinyan Liang [1]  Yuhua Qian [1]  Li Lv [1]

## Abstract

This paper studies a fundamental yet often overlooked premise in evolutionary multi-view classification (EMVC): the impact of label noise on EMVC, such as distorting fitness landscapes shaped by individual fitness values (e.g., test accuracy). Traditional EMVC assumes training labels are noise-free, yet this often fails in practice. As a result, label noise introduces harmful supervision during the training phase, resulting in distorted fitness landscapes and the emergence of fitness evaluation bias (FEB). This bias misguides the evolutionary trajectory, causing the search process to stagnate in local optima. Given that label noise largely stems from the mislabeling of samples near their decision boundaries by human annotators, we thus compared the decision boundaries of human annotators and models, and found discrepancies between the two. Based on this observation, we propose a simple yet effective "detect-then-calibrate" data purification framework that leverages outlier analysis in the gradient space (i.e., treating outliers as noisy samples) and prototype calibration in the feature space (i.e., utilizing feature prototypes of noise-free samples to correct the labels of noisy samples). Experimental results demonstrate that this strategy can effectively purify the data and alleviate FEB; moreover, it can improve the performance of various multi-view learning paradigms in label noise scenarios. (https://github.com/LiShuailzn/ICML-2026-GFD-EMVC)

[1]Institute of Big Data Science and Industry, Key Laboratory of Evolutionary Science Intelligence of Shanxi Province, School of Artificial Intelligence, Shanxi University. Correspondence to: Xinyan Liang <liangxinyan48@163.com>.

*Proceedings of the 43rd International Conference on Machine Learning*, Seoul, South Korea. PMLR 306, 2026. Copyright 2026 by the author(s).

## 1. Introduction

Multi-view classification (MVC) alleviates the insufficient expressive capability of single views by integrating complementary information of multiple views (Liang et al., 2022; He et al., 2026; Wei et al., 2025; Qin et al., 2026; Guo et al., 2024). However, existing MVC methods typically rely on manually pre-designed fusion structures and struggle to maintain favorable adaptability in complex and dynamic real-world scenarios. In recent years, the emerging evolutionary multi-view classification (EMVC) has introduced evolutionary algorithms into MVC (Liang et al., 2021), enabling the search process to dynamically adjust view fusion strategies according to the characteristics of multi-view data. It has demonstrated considerable effectiveness in various practical tasks, including protein secondary structure prediction (Qian et al., 2025), logic prediction (Guo et al., 2025). Fitness evaluation (FE) is the key driver of evolutionary search. It generates the ordering of individuals in the population by calculating fitness values (e.g., test accuracy) of individuals on the current task, and guides subsequent selection, crossover, and mutation operations accordingly. It is obvious that when fitness values fail to correctly characterize the true performance differences among individuals, the ordering of individuals will become disordered. As a result, the induced fitness landscapes is distorted, misguiding evolutionary trajectories and trapping them in local optima.

EFB-EMVC (Liang et al., 2025c) is the first work to study this issue from the perspective of view imbalance and term it as fitness evaluation bias (FEB). Different from the EFB-EMVC, this paper reveals another key cause of FEB from the more fundamental data level: harmfulness within data. Specifically, model training starts with data. When hard samples (e.g., samples difficult for humans to distinguish) exist in the data, they are more prone to mislabeling during the annotation process, thereby introducing label noise. These noisy samples continuously impose incorrect supervision during training, making it difficult for individuals to approximate their ideal performance. As shown in Fig. 1, this causes deviations in FE's estimation of the true capabilities of individuals, ultimately leading to FEB.

Although research on identifying potential noisy samples from raw data remains relatively scarce under the EMVC

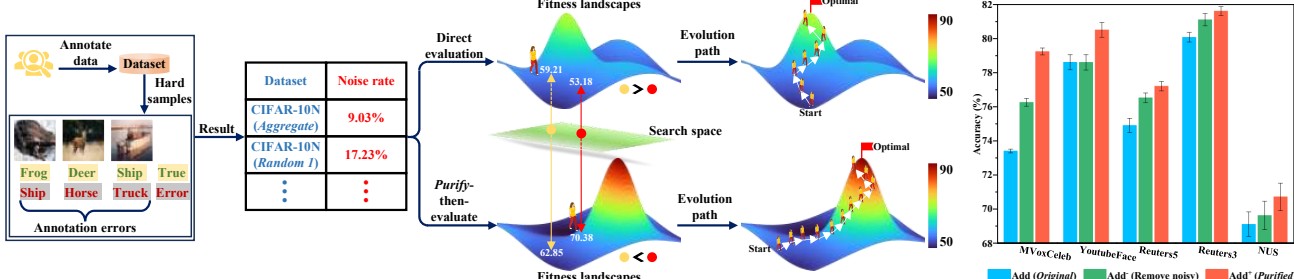

*Figure 1.* **Is the individual ranking derived from individual fitness values (e.g., test accuracy) calculated directly using given data reliable?** In real-world scenarios, hard samples (e.g., samples difficult for humans to distinguish) can induce annotation errors and introduce label noise; if individuals are directly evaluated on such data, noisy supervision will distort fitness landscapes and misguide evolutionary trajectories, trapping them in local optima. To this end, we enhance the reliability of fitness evaluation through a simple yet effective "detect-then-calibrate" data purification framework (i.e., using gradient space outlier analysis to identify noisy samples and leveraging feature space prototype calibration to correct their labels), thereby obtaining a more reasonable fitness landscapes and guiding a more accurate evolutionary trajectory. The results on the right demonstrate that: on multiple multi-view datasets, compared with *Original* (training on raw data), Remove noisy (training after removing noisy samples), and *Purified* (training after calibrating noisy samples), *Purified* consistently delivers more significant performance improvements.

framework, a large body of work has emerged in the field of label noise learning. They attempt to identify and filter potential noisy samples, and then train on relatively "clean" data subsets (Chhabra et al., 2025; Kim et al., 2024; Liang et al., 2025a). Among these, outlier gradient detection methods (Chhabra et al., 2025) point out that when noise such as label mislabeling exists in the data, such samples often exhibit gradient behavior inconsistent with that of most samples during the training process, thereby providing exploitable clues for characterizing data anomalies. Inspired by this, a natural assumption is that in EMVC, sample gradients generated during individual training can be uniformly mapped into the gradient space for outlier analysis to identify potential noisy samples, thus laying a more reliable data foundation for subsequent FE and evolutionary search.

However, noisy samples introduced by annotation errors are typically not randomly contaminated; instead, they are concentrated near the decision boundaries of human annotators. These samples are highly similar in appearance to adjacent classes, easily leading to inter-class misannotation (as shown on the left of Fig. 1, including Frog↔Ship, Deer↔Horse, and Ship↔Truck; where green denotes the ground truth and red indicates mislabels). In this sense, such samples reflect more of high inter-class similarity and discriminatively uncertain regions under human perception, rather than equating to purely "dirty data". Existing work (Chhabra et al., 2025) directly removes the corresponding samples upon detecting anomalies, and its implicit assumption is often that mislabeled samples near decision boundaries are difficult to calibrate reliably, and retaining them would instead introduce unstable training signals. However, this assumption does not always hold. We experimentally verify that the decision boundaries of human annotators and those of the model are not entirely consistent: because

models rely on high-dimensional feature representations to construct discriminative cues, which differ from those used by human annotators, some samples that are "difficult for humans to distinguish" may still be learnable and calibratable for models. Thus, such mislabeled samples are not useless noise caused by the distortion of the data itself, but rather have room for calibration; after calibration, they can even provide additional fine-grained supervision signals. Accordingly, directly removing samples upon detecting anomalous gradients may result in the erroneous elimination of a batch of high-value samples that are "difficult for humans to distinguish but learnable for models". In EMVC scenarios, this will further lead to the loss of complementary information provided by the same samples across multiple views, undermining the learning of cross-view complementary patterns.

Based on the above observations, we propose a simple yet effective "detect-then-calibrate" data purification framework. In the EMVC scenario, this framework examines the correlation between noisy samples and FEB from the data perspective, and designs a purification mechanism more aligned with the requirements of evolutionary search. Specifically, we first perform outlier analysis in the gradient space, treating outlier samples as noisy samples; we then conduct prototype calibration in the feature space, constructing feature prototypes using multi-view fused features of clean samples to rectify labels of noisy samples. Prior to the initiation of evolutionary search, it provides cleaner and more information-rich training data for various EMVC methods, thereby alleviating the FEB issue caused by label noise. Notably, our method is primarily targeted at "correctable mislabels": namely, samples that, although prone to confusion in human judgment, still possess separability under model representations. For mislabeled samples that also lie on the model decision boundary, such simple calibration may be

insufficient, and this aspect defines the scope of application of our method in this paper. The main contributions are:

- In the context of evolutionary multi-view classification, this work reveals and analyzes how label noise induce fitness evaluation bias (FEB) from a data perspective for the first time, enriching the research system of FEB.

- Experimental results reveal that differences exist between the decision boundaries of human annotators and those of the model. The finding induces a simple yet effective data purification via outlier analysis in the gradient space and prototype calibration in the feature space, alleviating FEB. Furthermore, calibrating samples lying on the decision boundaries of human annotators is superior to directly removing such samples.

- Experimental results demonstrate that the proposed method can not only effectively purify data and alleviate FEB, but also improve the performance of various multi-view learning paradigms in label noise scenarios.

## 2. Related Work

**Evolutionary Multi-View Classification (EMVC).** Unlike traditional multi-view classification that usually relies on human prior knowledge to construct fixed multi-view models (Liu et al., 2025; 2024; Shan et al., 2026; Zhong et al., 2025), EMVC transforms model structure design in multi-view learning into a search problem. It takes views and fusion operators as primitives to construct a search space, and leverages evolutionary algorithms to continuously generate, select, and refine candidate multi-view models within this space, thereby adaptively determining data fusion strategies suitable for the current task (Liang et al., 2021; 2025b;c). Most existing EMVC research focuses on improving search efficiency (Liang et al., 2024; Fu et al., 2024a;b; Cui et al., 2025). In contrast, the latest work (Liang et al., 2025c) for the first time identifies the phenomenon of fitness evaluation bias (FEB). Starting from the phenomenon of view imbalance, it introduces evolutionary navigators to explicitly guide the training of each view branch, achieving remarkable results. In contrast to this, this paper reveals another key inducement of FEB from a more fundamental data perspective: the harmfulness within data. Based on the experimental observation that "the decision boundaries of human annotators and those of the model are not entirely consistent", this paper proposes a simple yet effective "detect-then-calibrate" data purification framework to alleviate FEB.

**Label Noise Learning.** In the EMVC field, research on noisy samples is still lacking; in contrast, in the label noise learning field, a large number of works have attempted to identify noisy samples from the perspective of sample influence (Cook & Weisberg, 1982; Koh & Liang, 2017; Chhabra et al., 2025; Hammoudeh & Lowd, 2024). Existing methods mainly fall into two categories: retraining-based influence estimation (e.g., leave-one-out analysis (Cook & Weisberg, 1982)) and approximate influence estimation using gradients or second-order information (e.g., influence functions (Koh & Liang, 2017)). The former requires multiple model retrainings and has an unacceptable computational cost on deep models and large-scale datasets; the latter often relies on the assumption of the convexity of models and loss functions, leading to limited applicability to complex deep models. To address these issues, recent work (Chhabra et al., 2025) has turned to directly performing outlier detection in the gradient space to identify noisy samples.

However, most such methods adopt a "detect–remove" strategy. Under label noise, noisy samples are often located near the decision boundaries of human annotators; we find that the decision boundaries of human annotators and those of the model are not entirely consistent, which causes these samples to be prone to misannotation by human annotators but still separable under model representations. For this reason, this paper proposes a "detect-then-calibrate" data purification framework to alleviate FEB.

## 3. Proposed Method

The proposed data purification framework follows the basic workflow of traditional evolutionary multi-view classification (EMVC). Its key improvement lies in: purifying the given training set prior to the start of evolution, thereby alleviating the fitness evaluation bias (FEB) caused by label noise. The overall workflow is illustrated in Fig. 2.

### 3.1. Gradient and Feature Dual-Perception Framework

Based on the experimental observation that *the decision boundaries of human annotators and those of the model are not entirely consistent*, we propose a simple yet effective "detect-then-calibrate" data purification framework. It adopts a two-stage strategy of "gradient space detection + feature space calibration". The former serves to locate potential noisy samples, while the latter is used to correct their supervision signals. To avoid additional computational overhead from architecture search, we introduce an anchored model with a fixed architecture as a transfer model, which is used to provide gradient and feature representations.

#### 3.1.1. GRADIENT SPACE NOISE DETECTION

**Anchor model and gradient representation.** Let the given original dataset $D$ consist of $\mathcal{V}$ views. We construct an anchor model with a fixed structure, which adopts the simplest "full-view additive fusion" structure, and its binary

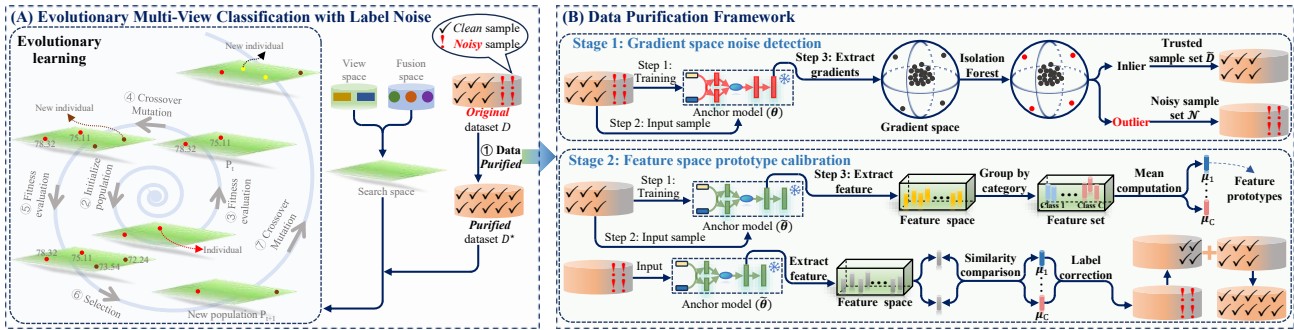

*Figure 2.* Evolutionary multi-view classification framework driven by data purification. (A) First, the original dataset $D$ is processed via the purification module to obtain purified dataset $D^\star$. Subsequently, the search space is constructed by means of view and fusion operators, and $D^\star$ drives the evolutionary search to obtain high-quality multi-view models. (B) The data purification framework adopts a two-stage strategy: Stage 1: Gradient space noise detection. The anchored model, trained on $D$ with parameters $\theta$, is used to extract sample gradients for constructing the gradient space. Outlier detection is then performed based on Isolation Forest, thereby dividing the samples into the trusted set $\widetilde{D}$ and the noisy set $\mathcal{N}$. Stage 2: Feature space prototype calibration. The anchored model is retrained (parameters $\tilde{\theta}$) based on the trusted set $\widetilde{D}$ to extract feature representations and construct the feature space, followed by the estimation of feature prototypes for each classes. Subsequently, features are extracted from samples in $\mathcal{N}$ and similarity matching is conducted with class prototypes to complete label calibration. Finally, these calibrated samples are merged with $\widetilde{D}$ to obtain the purified dataset $D^\star$.

tree encoding is

$$[\underbrace{v_1, v_2, \ldots, v_{\mathcal{V}}}_{\mathcal{V} \text{ elements}}, \underbrace{\text{Add}, \ldots, \text{Add}}_{\mathcal{V}-1 \text{ elements}}]. \quad (1)$$

We train the anchor model on $D$ until convergence to obtain parameters $\theta$. Subsequently, for each training sample $x_i$, we extract its gradient vector with respect to the parameters of the last linear layer:

$$\mathbf{g}_i = \nabla_{\theta_{\text{last}}} \ell(x_i, y_i; \theta), \quad (2)$$

where $\ell(\cdot)$ denotes the training loss, and $\theta_{\text{last}}$ represents the parameters of the last linear layer. Collecting the gradient vectors $\{\mathbf{g}_i\}$ of all samples enables the construction of a high-dimensional gradient space representation.

**Outlier detection and noisy set.** The noisy factors like labeling errors cause samples to exhibit abnormal gradient behaviors (Chhabra et al., 2025). iForest has advantages such as linear time complexity, low memory overhead, and suitability for high-dimensional gradient space modeling, thus being suitable for this scenario. Based on this observation, we adopt Isolation Forest (iForest) for outlier detection on $\{\mathbf{g}_i\}$ and set the outlier ratio to $\kappa\%$. Subsequently, outlier samples are regarded as potential noisy samples. Finally, we obtain the potential noisy sample set $\mathcal{N}$, and the remaining samples form the trusted set $\widetilde{D} = D \setminus \mathcal{N}$.

### 3.1.2. FEATURE SPACE PROTOTYPE CALIBRATION

**Obtaining a pure representation on the trusted set $\widetilde{D}$.** Since the noisy sample set $\mathcal{N}$ may contain noisy supervision, we retrain the anchored model on the $\widetilde{D}$ to construct more reliable class prototypes in the feature space, yielding

parameters $\tilde{\theta}$. Intuitively, the model trained on $\widetilde{D}$ is closer to "ideal supervision", and the intermediate representations it learns are more suitable for prototype construction and label calibration.

**Class prototypes and label calibration.** Load the parameters $\tilde{\theta}$ of the "pure" anchor model and extract the fused feature vectors $\mathbf{z}_i$ for samples in $\widetilde{D}$. Let there be $C$ classes; the class prototype for the $c$-th class is obtained by taking the mean of the feature vectors of samples in

$$\boldsymbol{\mu}_c = \frac{1}{\left|\widetilde{D}_c\right|} \sum_{(x_i, y_i) \in \widetilde{D}_c} \mathbf{z}_i, \quad c = 1, \ldots, C. \quad (3)$$

Subsequently, input the detected potential noisy samples $x_j \in \mathcal{N}$ into the anchor model (with parameters $\tilde{\theta}$) to extract their feature vectors $\mathbf{z}_j$, and calculate their cosine similarities with all class prototypes. The class with the maximum similarity is taken as the calibrated label:

$$\hat{y}_j = \arg \max_{c \in \{1, \ldots, C\}} \cos(\mathbf{z}_j, \boldsymbol{\mu}_c). \quad (4)$$

Finally, we merge the calibrated noisy samples with $\widetilde{D}$ to obtain the final purified dataset $D^\star$ for subsequent EMVC.

### 3.2. The EMVC Method Based on Purified Dataset

The procedure of the EMVC method based on the purified dataset proposed in this paper is presented in Algorithm 1. Given the original training set $D$, we first construct the purified dataset $D^\star$ via the gradient and feature dual-perception framework introduced in Section 3.1, and then use $D^\star$ as the training data for the subsequent evolutionary search phase.

Subsequently, we initialize the population in a search space composed of views and basic fusion operators. Specifically,

we randomly generate $k$ individuals to form the initial population $P$. Each individual is represented by a binary tree: leaf nodes denote views, and branch nodes denote basic fusion operators. The set of basic fusion operators is addition, multiplication, concatenation, maximize, average. For each individual in the population, we decode it into the corresponding multi-view model: first, each view is mapped to a unified fusion dimension $K$ via an encoder; then, a global representation is obtained through stepwise aggregation via the corresponding fusion operators following the predefined fusion order of the binary tree; finally, this global representation is fed into a classification head to output prediction results. We take the classification accuracy of the model on the test set as the fitness value of the individual. Based on fitness, the algorithm iteratively performs operations such as selection, crossover and mutation to update the population until the termination condition is met, and output a high-quality multi-view model.

Notably, $D^\star$ contains fewer noisy samples than $D$, so that fitness values obtained from training on $D^\star$ during the evolutionary process can more accurately reflect the true performance of individuals. This alleviates the FEB caused by label noise and guides the search to converge toward better solutions. More detailed implementation details of EMVC, including individual encoding and decoding, the implementation of selection, crossover and mutation, and other operations, can be found in Appendix A.8.

# 4. Experiments

We answer the following key questions for verifying the effectiveness of the proposed method. **Q1: Boundary Differences.** Do differences exist between the decision boundaries of human annotators and those of the model? **Q2: Purification Effectiveness.** Compared with various label noise learning baseline methods, can the proposed framework effectively identify and utilize noisy samples to achieve better learning performance? **Q3: FEB Existence & Alleviation, and Method Generalization.** When driving evolutionary multi-view classification (EMVC) with purified data, can the proposed framework reveal and alleviate the fitness evaluation bias (FEB) caused by label noise, and bring performance gains under different multi-view paradigms? **Q4: Parameter Sensitivity.** How do parameter choices affect the model's effectiveness? For visualization and ablation analysis of the dataset purification effect, see Appendix A.4.

## 4.1. Experimental Setup

To ensure experimental reproducibility and fair comparison, experiments related to label noise learning uniformly adopt ResNet-34 as the anchor model and are trained with the SGD optimizer (batch size=128, initial learning rate=0.1, momentum=0.9, weight decay=0.0005, trained for 100 epochs). Experiments related to EMVC use the Adam optimizer (learning rate=0.001, $\beta_1$=0.9, $\beta_2$=0.999); the clipping ratio $\kappa\%$ for outlier detection is fixed at 5% and 20% of the training data size for label noise learning and multi-view learning tasks respectively, and the number of trees in iForest is fixed at 100. More details are provided in Appendix A.1.

## 4.2. Analysis of Differences Between the Decision Boundaries of Human Annotators and Models

In the experiments, we take the *Aggregate* setting of CIFAR-10N as an example to verify that differences exist between the decision boundaries of human annotators and those of the model, thus providing strong evidence for the "simple yet effective" property of the method proposed in this paper. As illustrated in Fig. 3, human-mislabeled samples are not concentrated in the "inter-class transition regions" of the model representation space; instead, they are scattered within and around the clusters of various classes. This phenomenon indicates that "human mislabeling" does not necessarily correspond to "model inseparability", and further demonstrates that even with relatively simple strategies, samples not located near the model decision boundaries can be effectively corrected. More detailed experimental settings and results are provided in Appendix A.7.

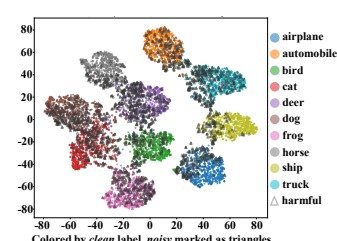

*Figure 3.* t-SNE visualization results of the CIFAR-10N (*Aggregate*) sample subset

## 4.3. Comparison with Label Noise Learning Methods

**Datasets.** We demonstrate the effectiveness of the proposed method in noisy sample identification and calibration on two real-world noisy label datasets, CIFAR-10N and CIFAR-100N (Wei et al., 2022). These datasets are derived from the original CIFAR-10 and CIFAR-100 datasets (Krizhevsky et al., 2009), but introduce label inaccuracies due to crowd-sourced annotation. CIFAR-10N has three different noise settings: *Aggregate*, *Random1*, and *Worst*—corresponding to the majority vote of three annotators, the label from the first annotator, and the label from the worst annotator, respectively. CIFAR-100N has only one noise setting. Their detailed descriptions can be found in Appendix A.2.

**Comparison Methods.** For label noise learning, baseline methods are grouped as follows. (1) Noisy-label correction methods: Normalized Margin (Northcutt et al., 2021), Self-Confidence (Müller & Markert, 2019), and Confidence-Weighted Entropy (Kuan & Mueller, 2022). (2) Influence-function-based methods: the Hessian-free gradient tracing

*Table 1.* Test accuracy (%, mean $\pm$ std, 5 runs) on CIFAR-10N and CIFAR-100N.

| Method | CIFAR-10N | | | CIFAR-100N |
|---|---|---|---|---|
| | Aggregate | Random | Worst | Noisy100 |
| Cross Entropy | 90.87 $\pm$ 0.23 | 89.17 $\pm$ 0.31 | 82.27 $\pm$ 0.37 | 57.36 $\pm$ 0.43 |
| Normalized Margin (JAIR 21) | 91.33 $\pm$ 0.11 | 90.06 $\pm$ 0.14 | 83.57 $\pm$ 0.32 | 60.94 $\pm$ 0.59 |
| Self-Confidence (IJCNN 19) | 91.38 $\pm$ 0.19 | 90.09 $\pm$ 0.17 | 83.65 $\pm$ 0.21 | 60.51 $\pm$ 0.51 |
| Confidence Entropy (ICML 22) | 91.11 $\pm$ 0.34 | 90.05 $\pm$ 0.26 | 83.63 $\pm$ 0.41 | 60.62 $\pm$ 0.26 |
| Gradient Tracing (NIPS 20) | 91.47 $\pm$ 0.21 | 89.98 $\pm$ 0.20 | 83.38 $\pm$ 0.58 | 60.73 $\pm$ 0.38 |
| LiSSA (ICML 17) | 91.49 $\pm$ 0.34 | 90.05 $\pm$ 0.31 | 83.38 $\pm$ 0.58 | 60.48 $\pm$ 0.29 |
| DataInf (ICLR 24) | 91.46 $\pm$ 0.17 | 90.05 $\pm$ 0.38 | 83.40 $\pm$ 0.56 | 60.70 $\pm$ 0.31 |
| Self-LiSSA | 92.07 $\pm$ 0.15 | 89.58 $\pm$ 0.11 | 83.01 $\pm$ 0.34 | 59.48 $\pm$ 0.43 |
| Self-DataInf | 91.41 $\pm$ 0.17 | 89.81 $\pm$ 0.37 | 83.15 $\pm$ 0.22 | 60.56 $\pm$ 0.28 |
| OGA (iForest) (ICML 25) | 91.36 $\pm$ 0.09 | 90.20 $\pm$ 0.07 | 83.72 $\pm$ 0.18 | 60.99 $\pm$ 0.27 |
| **GFD-EMVC (Ours)** | **92.08 $\pm$ 0.11** | **90.68 $\pm$ 0.12** | **84.46 $\pm$ 0.43** | **61.47 $\pm$ 0.29** |

approach of (Pruthi et al., 2020), LiSSA-based optimization (Koh & Liang, 2017), the recently proposed influence estimation method DataInf (Kwon et al., 2024), self-influence with LiSSA as in (Bejan et al., 2023), and self-influence with DataInf. Influence values are computed only for training samples, and performance is evaluated on the test set. (3) Outlier-gradient-based methods: Outlier Gradient Analysis (OGA) (Chhabra et al., 2025).

**Results.** The experimental results are presented in Table 1. Compared with training directly on noisy labels (Cross Entropy), OGA achieves a clear performance gain by removing noisy samples. However, due to its simple "detect-and-delete" strategy, it fails to fully exploit the useful information contained in noisy samples, leading to overall performance that is generally inferior to our proposed framework. Notably, under the two most severe noise settings—*Worst* (noise rate 40.21%) on CIFAR-10N and *Noisy100* (noise rate 40.20%) on CIFAR-100N—our method further surpasses the second-best approach by 0.74% and 0.48%, respectively. These results indicate that, after identifying noisy samples, calibrating their labels rather than discarding them can better leverage the useful information they contain. The results of all comparative methods in Table 1 are directly taken from (Chhabra et al., 2025). Overall, based on the experiments in Table 1 and the ablation experiments in Appendix A.4, the proposed framework can effectively identify and utilize noisy samples, achieving superior learning performance.

### 4.4. Experimental Results on Multi-view Learning

**Datasets.** In the experiments, five datasets are used, and they are MVoxCeleb (Liang et al., 2025b), YoutubeFace (Wang et al., 2022), NUS-WIDE-128 (NUS) (Tang et al., 2017) and Reuters (Amini et al., 2009). For the Reuters dataset, two variants named Reuters5 and Reuters3 are generated

by adding Gaussian noise (Liang et al., 2025b). Since these datasets do not originally contain noisy labels, we artificially inject label noise into the training set: we randomly select 40% of the training samples and flip their labels uniformly at random to a different class, thereby constructing noisy versions of the datasets. The test set remains clean. Their detailed descriptions can be found in Appendix A.2.

**Comparison Methods.** To validate the effectiveness of our proposed data purification framework, we train three types of multi-view classification methods on the original dataset and on the purified dataset, respectively. Specifically: (1) Fixed fusion methods, including addition, average, max, multiplication, and concatenation; (2) Trustworthy multiview classification methods, including ETMC (Han et al., 2023), RCML (Xu et al., 2024), and ETF (Lu et al., 2025); (3) EMVC methods, including EDF (Liang et al., 2021), DC-NAS (Liang et al., 2024) and EFB-EMVC (Liang et al., 2025c). Note that DC-NAS extends EDF with two key extensions: a) replacing sequential individual encoding with binary tree encoding; b) using a data divide-and-conquer strategy to speed up the search process, yet it typically sacrifices some performance. In our experiments, we focus on classification performance rather than search speed. Thus, in implementing DC-NAS, we retain its binary tree encoding, use full data and omit the data divide-and-conquer step to maximize its performance upper bound.

#### 4.4.1. ANALYSIS OF THE EXISTENCE OF FEB

In this experiments, we aim to verify that label noise in multi-view datasets can induce FEB. Specifically, we randomly initialize a population consisting of 15 individuals on the first fold of data from the YoutubeFace dataset (the individuals used are provided in detail in Appendix A.3), train the individuals on the original training set and the

*Table 2.* Rankings and Correlation Coefficients (PC: Pearson Correlation; SRC: Spearman's Rank Correlation; KT: Kendall's Tau).

| Dataset | Sort the fitness values of 15 individuals | PC | SRC | KT |
|---|---|---|---|---|
| *Original* dataset | 1, 10, 9, 13, 14, 8, 11, 4, 5, 7, 2, 6, 12, 3, 15 | 0.3857 | 0.3857 | 0.2571 |
| *Purified* dataset | 1, 14, 4, 10, 9, 11, 8, 13, 7, 5, 2, 6, 3, 12, 15 | | | |

*Table 3.* Pairs of individuals with changed fitness value relationships between *Original* and *Purified* dataset.

| Pair | Individual Pair | *Original* Dataset | *Purified* Dataset |
|---|---|---|---|
| (1) | [0, 3, 1] vs [4, 0, 3] | 53.18 vs 59.21 | 70.38 vs 62.85 |
| (2) | [1, 0, 4, 2, 0, 2, 0] vs [1, 4, 2] | 73.05 vs 73.38 | 76.78 vs 76.56 |
| (3) | [1, 0, 4, 2, 0, 2, 0] vs [0, 1, 3, 3, 4] | 73.05 vs 75.50 | 76.78 vs 76.59 |
| (4) | [1, 0, 4, 2, 0, 2, 0] vs [1, 3, 4, 1, 0] | 73.05 vs 76.04 | 76.78 vs 76.68 |
| (5) | [1, 0, 4, 2, 0, 2, 0] vs [4, 1, 2] | 73.05 vs 73.37 | 76.78 vs 76.58 |
| (6) | [1, 0, 4, 2, 0, 2, 0] vs [2, 1, 0, 3, 1, 0, 4] | 73.05 vs 75.05 | 76.78 vs 75.75 |
| (7) | [1, 2, 3] vs [3, 0, 2, 4, 1, 0, 1] | 71.46 vs 71.17 | 73.50 vs 75.74 |
| (8) | [1, 4, 2] vs [4, 1, 2] | 73.38 vs 73.37 | 76.56 vs 76.58 |
| (9) | [1, 4, 2] vs [2, 1, 0, 3, 1, 0, 4] | 73.38 vs 75.05 | 76.56 vs 75.75 |
| (10) | [0, 1, 3, 3, 4] vs [0, 4, 3, 2, 1, 1, 0, 4, 2] | 75.50 vs 74.84 | 76.59 vs 77.69 |
| (11) | [1, 3, 4, 1, 0] vs [0, 4, 3, 2, 1, 1, 0, 4, 2] | 76.04 vs 74.84 | 76.68 vs 77.69 |
| (12) | [4, 1, 2] vs [2, 1, 0, 3, 1, 0, 4] | 73.37 vs 75.05 | 76.58 vs 75.75 |
| (13) | [2, 1, 0, 3, 1, 0, 4] vs [0, 4, 3, 2, 1, 1, 0, 4, 2] | 75.05 vs 74.84 | 75.75 vs 77.69 |

purified training set respectively, and sort them according to the final fitness values. Results are shown in Table 2, where the correlation coefficient between the two orderings is low. Furthermore, we count the ordering inversions as presented in Table 3; notably, 13 pairs of individuals with inverted orderings are observed in a population with only 15 individuals, indicating that label noise can induce FEB. In addition, this section validates the existence of FEB using only static experimental results; however, static metrics do not clearly reflect how label noise gradually misleads the evolutionary direction. To improve intuition, Appendix A.9 further provides evidence from the dynamic evolution process, presenting FEB from a process-level perspective.

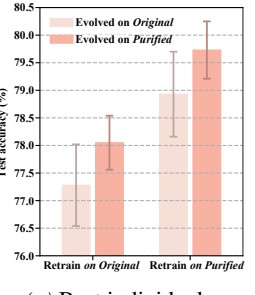

*(a)* Best individuals.

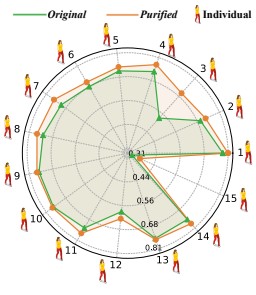

*(b)* Population.

*Figure 4.* Test accuracy comparison on *Original* and *Purified* datasets: (a) individual test accuracy (%) under the same dataset conditions; (b) individuals test accuracy in the population.

### 4.4.2. ANALYSIS OF ALLEVIATION EFFECT

In this experiments, we aim to verify whether the proposed method can effectively alleviate FEB. Table 4 demonstrates that the performance of all EMVC methods is significantly improved after training on the purified dataset. To rule out the possibility that the observed accuracy improvement stems merely from cleaner data rather than the algorithm converging to better solutions, we further verify whether data purification drives the algorithm to converge to higher-quality individuals. Specifically, under the 5-fold cross-validation partition of the YoutubeFace dataset, we extract the optimal individuals evolved on the original dataset and the purified dataset respectively (details of the optimal individuals are provided in Appendix A.6), and then retrain and evaluate both types of individuals on the original and purified datasets. If, under the same training dataset conditions, the optimal individuals evolved on the purified dataset outperform those evolved on the original dataset, it can be con-

firmed that the purified dataset drives the algorithm to converge to better solutions. As presented in Fig. 4a, whether trained on the original dataset or the purified dataset, the optimal individuals evolved on the purified dataset achieve higher test accuracy. This indicates that data purification not only alters the evolutionary trajectory but also guides the algorithm to converge to better solutions, thereby alleviating FEB efficiently. Finally, Fig. 4b further shows that the performance of all individuals in the population is generally improved after data purification.

In summary, the proposed method can reveal and alleviate FEB caused by label noise.

### 4.4.3. GENERALITY ANALYSIS

In this experiments, we aim to verify whether the proposed method can consistently yield performance gains across dif-

*Table 4. Method* denotes training on the original dataset, and *Method*$^+$ denotes training on the dataset purified by our proposed method.

| Groups | Methods | MVoxCeleb | YoutubeFace | Reuters5 | Reuters3 | NUS |
|---|---|---|---|---|---|---|
| Fixed fusion methods | Add | 70.23±0.24 | 76.51±0.41 | 75.07±0.47 | 79.98±0.28 | 67.39±0.90 |
| | Add$^+$ | **76.93±0.10** | **78.52±0.26** | **77.70±0.34** | **81.74±0.13** | **70.47±0.82** |
| | Mul | 61.80±0.49 | 76.20±0.27 | 28.15±1.98 | 37.04±2.98 | 26.76±11.12 |
| | Mul$^+$ | **74.38±0.24** | **79.68±0.18** | **69.46±2.84** | **80.21±0.61** | **62.53±1.22** |
| | Cat | 53.13±0.41 | 75.02±0.26 | 73.79±0.23 | 79.05±0.61 | 54.96±1.90 |
| | Cat$^+$ | **64.51±0.15** | **78.56±0.49** | **75.79±0.52** | **81.44±0.22** | **62.38±1.00** |
| | Max | 60.14±0.18 | 74.43±1.22 | 73.20±0.26 | 78.68±0.37 | 66.12±0.42 |
| | Max$^+$ | **68.46±0.40** | **77.99±0.29** | **77.71±0.41** | **81.83±0.14** | **69.54±0.93** |
| | Avg | 70.44±0.29 | 74.24±1.14 | 76.77±0.50 | 81.20±0.34 | 68.30±0.92 |
| | Avg$^+$ | **77.01±0.27** | **76.88±0.64** | **78.49±0.38** | **82.38±0.31** | **70.49±1.05** |
| Trusted MVC methods | ETMC (TPAMI23) | 56.22±0.65 | 52.23±1.50 | 74.53±0.13 | 79.35±0.22 | 56.64±0.71 |
| | ETMC$^+$ | **65.46±0.28** | **70.96±0.75** | **75.87±0.38** | **80.61±0.33** | **63.21±0.65** |
| | RCML (AAAI24) | 46.04±0.18 | 34.48±1.46 | 78.19±0.14 | 81.89±0.29 | 64.84±0.64 |
| | RCML$^+$ | **56.78±0.12** | **45.33±2.41** | **78.59±0.31** | **82.32±0.22** | **67.74±0.54** |
| | ETF (ICML25) | 75.96±0.43 | 73.03±0.38 | 76.43±0.48 | 80.94±0.55 | 67.81±0.87 |
| | ETF$^+$ | **80.09±0.18** | **78.76±0.20** | **77.82±0.45** | **81.31±0.64** | **70.36±0.96** |
| EMVC methods | EDF (TEVC2021) | 70.93±0.30 | 78.89±0.52 | 77.24±0.30 | 81.90±0.38 | 68.95±0.63 |
| | EDF$^+$ | **77.26±0.15** | **80.91±0.31** | **78.92±0.31** | **82.75±0.33** | **71.17±0.89** |
| | DC-NAS (AAAI24) | 70.91±0.20 | 78.90±0.58 | 77.18±0.20 | 81.67±0.26 | 69.07±0.53 |
| | DC-NAS$^+$ | **77.39±0.25** | **80.90±0.42** | **78.86±0.25** | **82.75±0.24** | **71.28±0.58** |
| | EFB-EMVC (NIPS25) | 75.06±0.13 | 81.26±0.29 | 77.08±0.23 | 81.90±0.33 | 70.11±0.50 |
| | EFB-EMVC$^+$ | **80.55±0.16** | **82.05±0.49** | **78.60±0.27** | **82.74±0.34** | **71.78±0.69** |

ferent multi-view learning paradigms. Experimental results are shown in Table 4. By comparing the test accuracy on the original noisy datasets and on the datasets purified by our proposed method, we observe that all multi-view learning methods achieve significant performance improvements on each dataset, which clearly validates the generality of the proposed data purification approach. In summary, our proposed method consistently yields performance gains across different multi-view learning paradigms.

### 4.5. Parameter Sensitivity Analysis

In this experiment, we use the *Aggregate* setting of CIFAR-10N to analyze the sensitivity of the proposed method to key parameters. We vary the clipping ratio $\kappa\%$ over [2.5%, 5%, 7.5%, 10%, 12.5%], with results reported in Table 5.

*Table 5.* Test accuracy (%) on CIFAR-10N (*Aggregate*) under different clipping ratios $\kappa\%$.

| $\kappa\%$ | 2.5% | 5% | 7.5% | 10% | 12.5% |
|---|---|---|---|---|---|
| Test accuracy | 91.64 | 92.36 | 92.38 | 92.21 | 92.17 |

It can be observed that: when $\kappa\%$ is below the actual noise rate of the dataset (9.03%), the accuracy of the model rises gradually with the increase of $\kappa\%$, indicating that moderate calibration of noisy samples can effectively improve performance; when $\kappa\%$ exceeds the noise rate, the accuracy begins to decline, suggesting that excessive calibration harms normal samples and thus impairs model performance. In practical scenarios, the true noise rate of the dataset is often unknown in advance. To address this issue, we recommend adopting 5% as the initial value of $\kappa\%$, while the ideal value of $\kappa\%$ needs to be tuned according to specific tasks. Besides $\kappa\%$, sensitivity analysis results for other parameters are detailed in Appendix A.5.

## 5. Conclusion

In the evolutionary multi-view classification (EMVC), we reveal and analyze for the first time from the data level how label noise induces fitness evaluation bias (FEB). Based on the experimental observation that differences exist between the decision boundaries of human annotators and those of the model, we propose a simple yet effective "detect-then-calibrate" data purification framework, thereby alleviating FEB caused by label noise. Furthermore, applying the purified data to various multi-view learning methods leads to significant performance improvements, which demonstrates the generality of the proposed method. However, it should be noted that the proposed method struggles to achieve reliable calibration for noisy samples located near model decision boundaries.

Looking forward, there are still several key issues worth in-depth research. For example, what other potential factors may induce FEB beyond the data perspective, and how to design the corresponding alleviation mechanisms? Do there exist more superior data purification strategies that can effectively calibrate the noisy samples at the model decision boundary? These issues will all become key research directions to promote the sustainable development of EMVC.

## Acknowledgements

This work was supported by National Natural Science Foundation of China (Nos. 62306171, T2495251, 62136005) and the Special Fund for Science and Technology Innovation Teams of Shanxi Province (No. 202304051001001).

## Impact Statement

This paper presents work whose goal is to advance the field of Machine Learning. There are many potential societal consequences of our work, none which we feel must be specifically highlighted here.

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

# A. Appendix

In the supplemental material:

## A.1. Experimental Settings

### A.1.1. EXPERIMENTAL ENVIRONMENT

The computing environment includes Ubuntu 24.04.2 LTS as the operating system, equipped with an AMD EPYC processor with 160 physical cores (320 logical threads), 566 GB of DDR4 memory, and 8 NVIDIA GeForce RTX 5090 GPUs, each with 32 GB of VRAM. In our experiments,

- For the label noise learning comparison experiments, to ensure consistency with existing methods, we implement all experiments using the PyTorch framework (PyTorch 2.8.0+cu128, Python 3.10.18, CUDA 12.8).

- For the multi-view learning comparison experiments, we implement all experiments using TensorFlow framework (TensorFlow 2.10.0 GPU, Python 3.9.23, CUDA 11.2).

### A.1.2. PARAMETER SETTINGS

#### (a) Compared Experiments with Label Noise Learning Methods

In Table 1 experiments, to ensure fairness, we uniformly adopt ResNet-34 as the anchor model, which is used in the literature (Chhabra et al., 2025). The training configuration is: batch size 128, SGD optimizer, initial learning rate 0.1, momentum 0.9, weight decay 0.0005, and 100 training epochs. It should be noted that we directly used the implementation provided in (Chhabra et al., 2025) and made modifications based on their code. The clipping ratio $\kappa\%$ for outlier detection is uniformly set to 5% of the training data size. In addition, among the commonly used benchmark datasets in label noise learning (specifically including CIFAR-10N and CIFAR-100N), after extracting the gradients of the last layer of ResNet-34, we adopt Sparse Random Projection (Li et al., 2006) technology for dimensionality reduction. The projection dimensions of the gradients under each noise setting are as follows: 1) *Aggregate*: 2048; 2) *Random*: 1024; 3) *Worst*: no projection; 4) *Noisy100*: 8192. The number of trees in iForest uses the default value of 100.

#### (b) Experimental Results on Multi-View Learning

Among commonly used multi-view datasets (specifically including MVoxCeleb, YoutubeFace, Reuters5, Reuters3 and NUS), when extracting the gradients of the last layer of anchored model, due to the extremely high dimension of the gradient vectors corresponding to the MVoxCeleb dataset, we adopt Sparse Random Projection for this dataset to reduce the gradient dimension, where the target dimension is adaptively determined by the algorithm. For the other datasets, outlier detection is performed directly in the original gradient space without additional projection. The clipping ratio $\kappa\%$ for outlier detection is uniformly set to 20% of the training data size. The number of trees in iForest uses the default value of 100.

In the experiments of Table 4, the hyperparameter settings for the three categories of multi-view learning methods (Fixed fusion methods, Trusted MVC methods, and EMVC methods) are as follows:

*(1) Fixed fusion methods*: The dimension of the fused view is uniformly set to 128, and it is uniformly encoded in the form of sequences.

*(2) Trusted MVC methods*: 1) ETMC: batch_size=512, learning rate=1e-3, training epochs=500; 2) RCML: batch_size=512, learning rate=1e-3, training epochs=200, gamma=1.0; 3) ETF: batch_size=200, training epochs=500, warm-up epochs=1, smoothing coefficient=0.9; learning rates are set as follows: lr=3e-4, rlr=1e-3 on the MVoxCeleb dataset; lr=1e-3, rlr=3e-4 on the YoutubeFace dataset; lr=1e-4, rlr=3e-4 on the Reuters5 dataset; lr=3e-3, rlr=1e-4 on the Reuters3 dataset; lr=1e-3, rlr=1e-3 on the NUS dataset.

*(3) EMVC methods*: All MVM are trained using the Adam algorithm. The learning rate is set to 0.001, with a first-moment exponential decay rate of 0.9 and a second-moment exponential decay rate of 0.999. Each network undergoes training for 200 epochs. To prevent overfitting, if the performance of a MVM does not improve after 10 epochs, the training process will be halted. Inspired by the work of EDF (Liang et al., 2021), we set the population size to 28, the number of iteration rounds to 20, the crossover rate to 0.9, and the mutation rate to 0.2. The fusion view dimension to 128 and the reuse of view features is not allowed. EDF is encoded in the form of sequences, while DC-NAS and EFB-EMVC are encoded in the form of binary trees.

### A.1.3. Evaluation Protocol

To reduce the randomness caused by data partitioning and network initialization, five-fold cross-validation is adopted for multi-view datasets (MVoxCeleb, YouTubeFace, Reuters5, Reuters3, NUS); for noisy label datasets (CIFAR-10N, CIFAR-100N), since the original work has provided fixed data partitioning, we conduct five independent repeated experiments under this partitioning. All results are reported in the form of "mean ± standard deviation".

### A.2. Datasets

- **MVoxCeleb** (Liang et al., 2025b) is a multi-view audio classification dataset that is constructed with VoxCeleb dataset (Nagrani et al., 2020). Each audio are extracted five view features and they are two deep feature including ecapa and resnet, and three traditional features including fbank, mfcc and spec. To study aim, Gaussian noise is added on ecapa and resnet mfcc.

- **YouTube-Faces** (Wang et al., 2022). The dataset includes 3,425 videos of 1,595 different people downloaded from YouTube. Similar to, we use a subset consists of 101,499 frames of 31 subjects and the same five features are extracted.

- **NUS-WIDE-128 (NUS)** (Tang et al., 2017). NUS dataset contains 43,800 single label images from 128 categories. For each image, six types of image features including color histogram (CH), color correlogram (CORR), edge direction histogram (EDH), wavelet texture (WT), block-wise color moments (CM) and bag of words based on SIFT descriptions (BoW), and one text feature are extracted. The dataset extended from the NUS-WIDE dataset (Chua et al., 2009). In our experiments, we use its a subset consisting of 23,438 images from 10 category, including *animal*, *architecture*, *art*, *flowers*, *food*, *man*, *person*, *sky*, *toy*, and *water*. In this subset, each image is related to one label and each category includes at least 1,500 images.

- **Reuters** (Amini et al., 2009). Reuters is a multilingual multi-view dataset, each document is described by five different languages including English, French, German, Spanish and Italian. To make used model can work on this data, the dimensions of all views are reduced to 1,000 using PCA. Then, following, Gaussian noise is added to all views or 3 views for obtaining its two versions, named as Reuters5 (Liang et al., 2025b) and Reuters3 (Liang et al., 2025b), respectively.

- **CIFAR-10N AND CIFAR-100N** (Wei et al., 2022). Both of them are label-noise versions constructed on top of the classical datasets CIFAR-10 and CIFAR-100 (Krizhevsky et al., 2009). They share exactly the same image inputs as the original datasets: each sample is an RGB image of size $3 \times 32 \times 32$, and the numbers of classes are 10 and 100, respectively. In contrast to the original versions, training labels of CIFAR-10N/100N are independently provided by three annotators on Amazon Mechanical Turk, and therefore inevitably contain real human annotation errors. Compared with commonly used synthetic noise settings, this type of noise induced by human annotation is closer to label-noise scenarios in practical applications. The scales of the two datasets are consistent with the original CIFAR: each training set contains 50,000 image–label pairs, and each test set contains 10,000 samples with clean labels. For CIFAR-10N, three official noise configurations are used in our experiments: *Worst*: the version with the highest noise rate (40.21%),

which corresponds to labels given by the worst annotator among the three; *Aggregate*: aggregated labels obtained by majority voting over the three annotators, with an overall noise rate of 9.03%; *Random*: labels randomly selected from one of the three annotators (fixed as the first annotator in this paper), with a noise rate of about 17.23%. For the CIFAR-100N dataset, we use CIFAR-100N Fine (denoted as *Noisy100*), with an overall noise rate of 40.20% for this version.

In this paper, based on the experimental observation that differences exist between the decision boundaries of human annotators and those of the model, we propose a "detect-then-calibrate" data purification framework. The effectiveness of the proposed method has been validated on datasets containing real-world label noise, specifically CIFAR-10N and CIFAR-100N. It should be noted that there is a lack of real noisy versions with human re-annotation records analogous to CIFAR-10N/100N for multi-view datasets at present. The multi-view data utilized in this study consists of publicly released pre-extracted features, without access to raw samples and annotation process information. Thus, it is impossible to reliably estimate which categories are more confusable for human annotators, and further inject noise that aligns more closely with real annotation behavior. In addition, manual re-annotation for multiple multi-view datasets is not feasible in terms of cost. Given the above objective constraints, we adopt the most common random flipping strategy in the label noise learning field to construct noisy versions for multi-view experiments, which are used to evaluate the effectiveness of the proposed method in multi-view scenarios under standard reproducible settings. We also leave the construction of real noisy multi-view benchmark datasets as future work.

### A.3. Individuals Used in FEB Experiment

In this section, the 15 individuals used in the FEB experiments are listed in detail in Table 6. In this section, we adopt a sequence-based individual structure for encoding and represent each individual in the form of [(*view*), (*fusion operator*)]. If an individual contains $\mathcal{V}$ views, it must contain $\mathcal{V} - 1$ fusion operators to ensure that all views included in the individual are ultimately aggregated into a single global representation.

*Table 6.* The set of individuals used in the FEB experiment

| Serial Number | Individual |
|:---:|:---:|
| (1) | [ (0, 3, 1, 2, 4), (1, 2, 4, 1) ] |
| (2) | [ (1, 2), (4) ] |
| (3) | [ (0, 3), (1) ] |
| (4) | [ (1, 0, 4, 2), (0, 2, 0) ] |
| (5) | [ (1, 2), (3) ] |
| (6) | [ (3, 4), (4) ] |
| (7) | [ (3, 0, 2, 4), (1, 0, 1) ] |
| (8) | [ (1, 4), (2) ] |
| (9) | [ (0, 1, 3), (3, 4) ] |
| (10) | [ (1, 3, 4), (1, 0) ] |
| (11) | [ (4, 1), (2) ] |
| (12) | [ (4, 0), (3) ] |
| (13) | [ (2, 1, 0, 3), (1, 0, 4) ] |
| (14) | [ (0, 4, 3, 2, 1), (1, 0, 4, 2) ] |
| (15) | [ (2, 0), (0) ] |

### A.4. Visualization and Ablation Analysis of Dataset Purification Effect.

In this experiments, we conduct a visualization-based ablation study on the key steps of our framework, focusing on two questions: (1) whether the samples detected as outliers are indeed harmful, and (2) whether their true classes can be recovered after label calibration. To this end, we take the *Aggregate* setting of CIFAR-10N as an example and visually analyze the detected outlier samples and their calibrated results to further verify the rationality and effectiveness of the proposed method.

The visualization results in Fig. 5 show that: (1) the samples detected as outliers by our method exhibit labels that are clearly

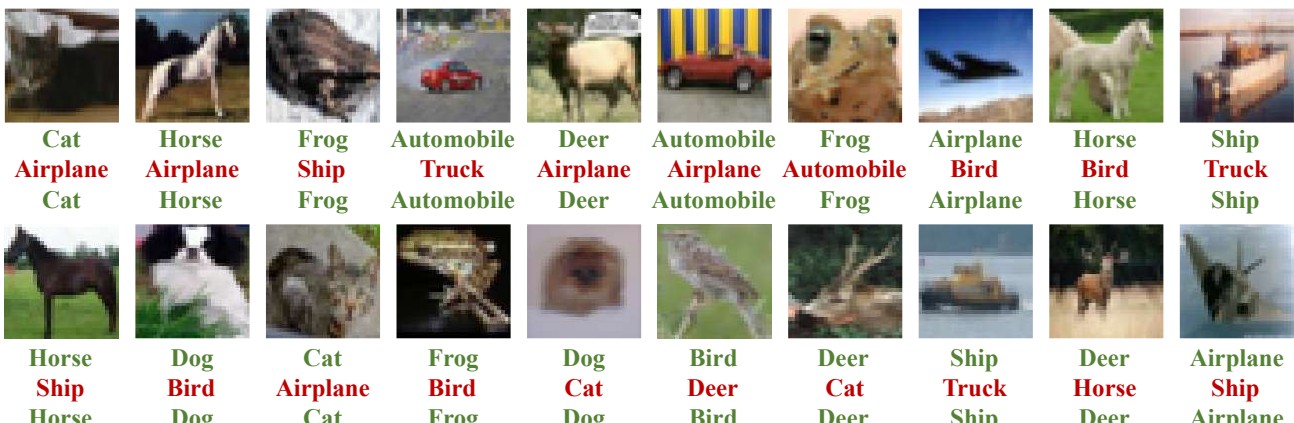

*Figure 5.* The three lines of text below the figure indicate: ***First line***: true labels in the original CIFAR-10; ***Second line***: noisy labels in CIFAR-10N; ***Third line***: the labels after our method detects and calibrates the noisy samples.

inconsistent with the true classes implied by their image content, indicating that the identified samples are indeed noisy; and (2) after applying label calibration, these noisy samples are successfully corrected to the true classes consistent with the image content, demonstrating the effectiveness of the proposed label calibration strategy.

To further verify the effectiveness of the proposed method in multi-view scenarios, we additionally conducted ablation experiments on multiple multi-view datasets and set the outlier detection ratio to 20%. We compared the test accuracy under three anchored model training strategies: 1) Training directly on the original noisy dataset; 2) Training after directly removing the noisy samples detected by the method; 3) Training on the dataset purified by the method. It should be noted that although the anchored models also adopt a full-view additive architecture, their specific implementation is not entirely consistent with the additive architecture in Table 4. On the one hand, the anchored models employ a binary tree structure for encoding, whereas the architecture in Table 4 relies on sequential addition. On the other hand, the anchored models additionally incorporate more sophisticated designs such as dropout into each view branch, aiming to achieve more desirable data purification performance through a more powerful model structure, while such operations are not included in the additive architecture in Table 4.

As shown in Table 7, directly removing detected noisy samples can bring about a certain degree of performance improvement in most cases, but further performing label calibration on this basis usually achieves higher test accuracy. This indicates that merely removing noisy samples may lead to the loss of effective information, while the proposed label calibration strategy can suppress noise interference while preserving and utilizing effective supervision signals, thereby further improving model performance.

*Table 7.* Test accuracy (%) comparison of different training strategies on multi-view datasets. *Methods*: Training on the original noisy dataset; *Methods$^-$*: Training after removing detected noisy samples; *Methods$^+$*: Training on the purified dataset.

| Methods | MVoxCeleb | YoutubeFace | Reuters5 | Reuters3 | NUS |
|---------|-----------|-------------|----------|----------|-----|
| *Anchor model* | 73.41±0.11 | 78.60±0.44 | 74.91±0.42 | 80.06±0.28 | 69.13±0.72 |
| *Anchor model$^-$* | 76.26±0.23 | 78.59±0.45 | 76.52±0.28 | 81.09±0.36 | 69.64±0.83 |
| *Anchor model$^+$* | **79.23±0.20** | **80.49±0.43** | **77.20±0.27** | **81.59±0.26** | **70.73±0.80** |

### A.5. Parameter Sensitivity Analysis

In this experiment, we take the *Aggregate* setting of CIFAR-10N as an example to conduct a sensitivity analysis on key parameters of the proposed method except the clipping ratio $\kappa\%$, aiming to evaluate the robustness of the model under different parameter choices. Specifically, the analyzed parameters include: (1) the target projection dimension in the high-dimensional gradient space; (2) the number of trees in iForest.

A.5.1. TARGET PROJECTION DIMENSION OF THE HIGH-DIMENSIONAL GRADIENT SPACE

In the experiment, the projection dimension of the high-dimensional gradient space is sequentially set to [256, 512, 1024, 2048, 4096], and the corresponding results are shown in Table 8. It can be observed that within this range, changes in the projection dimension have little impact on model test accuracy, indicating that the proposed method generally exhibits good robustness to this parameter. Based on the above experimental results, we recommend adopting 256 as the initial value of the projection dimension and optimizing it according to specific tasks.

*Table 8.* Test accuracy (%) under different projection dimensions (PD) of CIFAR-10N (*Aggregate*)

| PD | 256 | 512 | 1024 | 2048 | 4096 |
|---|---|---|---|---|---|
| Test accuracy | 92.12 | 92.05 | 92.19 | 92.21 | 92.16 |

A.5.2. NUMBER OF TREES IN IFOREST

In the experiment, we sequentially set the number of trees in iForest to [50, 75, 100, 125, 150], with the corresponding results presented in Table 9. Overall, as the number of trees increases, the range of performance fluctuations remains limited, indicating that the proposed method also exhibits good robustness to this parameter in general. Therefore, without additional parameter tuning, we recommend setting the number of trees to 100 as the default configuration, which is also the setting consistently used across all our experiments.

*Table 9.* Test accuracy (%) under different numbers of trees of CIFAR-10N (*Aggregate*).

| Trees | 50 | 75 | 100 | 125 | 150 |
|---|---|---|---|---|---|
| Test accuracy | 92.10 | 92.34 | 91.67 | 92.06 | 91.98 |

## A.6. Demonstration of the Best Individuals Obtained by EDF on YouTubeFace under Two Data Settings

In this section, the optimal individual encoding structures of the EDF method obtained through iteration based on the original dataset and the purified dataset, respectively, under the five-fold cross-validation setup on the YoutubeFace dataset are presented in detail, with the results shown in Table 10. In this section, the sequence-form individual encoding structure is also adopted.

*Table 10.* Optimal individual encoding structures obtained through iteration under the two data conditions (*Original*/*Purified*), respectively, under the 5-fold cross-validation setup on the YoutubeFace dataset

| YoutubeFace | Individual Source | Individual |
|---|---|---|
| Fold 1 | Evolved on *Original* dataset | [ (3, 1, 4), (2, 1) ] |
| | Evolved on *Purified* dataset | [ (3, 1, 2, 0, 4), (0, 1, 2, 1) ] |
| Fold 2 | Evolved on *Original* dataset | [ (4, 1, 3, 2), (1, 3, 0)] |
| | Evolved on *Purified* dataset | [ (3, 0, 1, 2), (2, 4, 1)] |
| Fold 3 | Evolved on *Original* dataset | [ (3, 0, 1, 2), (2, 4, 4)] |
| | Evolved on *Purified* dataset | [ (1, 3, 2, 4), (2, 1, 1)] |
| Fold 4 | Evolved on *Original* dataset | [ (0, 3, 1, 2), (2, 4, 3) ] |
| | Evolved on *Purified* dataset | [ (2, 3, 1), (2, 1) ] |
| Fold 5 | Evolved on *Original* dataset | [ (3, 1 ,2, 4), (2, 0, 4) ] |
| | Evolved on *Purified* dataset | [ (0, 1, 3, 4, 2), (0, 2, 4, 1) ] |

## A.7. Analysis of Differences Between the Decision Boundaries of Human Annotators and Models

In this experiments, we present evidence for the fact that the decision boundaries of human annotators and those of the model are not entirely consistent from a visualization perspective, thereby explaining why simple label calibration on mislabeled

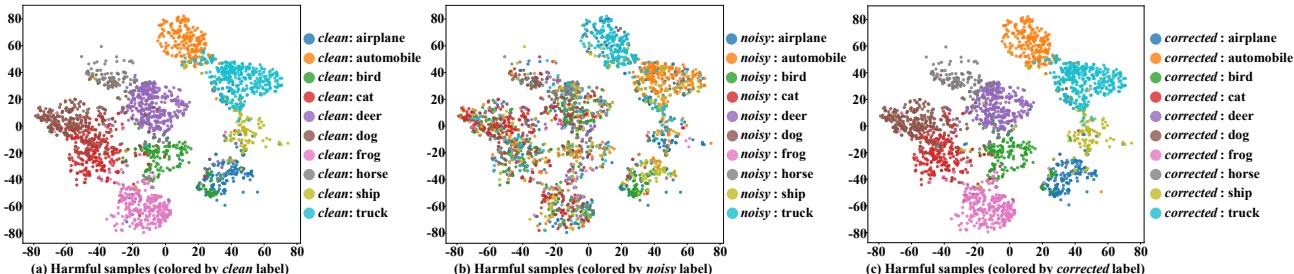

*Figure 6.* (a) Color the detected harmful samples according to their original true labels; (b) Color the detected harmful samples according to their noisy labels; (c) Color the detected harmful samples according to the labels corrected by the proposed method.

samples yields significant gains. We take the *Aggregate* setting of CIFAR-10N as an example and fix the anchor model structure as ResNet-34 (parameters trained on the given original dataset $D$ are denoted as $\theta$; parameters trained on the trusted dataset $\widetilde{D}$ after removing harmful samples are denoted as $\tilde{\theta}$). For all visualization experiments in this section, we extract the feature vectors preceding the classifier head of the anchored models as the feature representation.

### A.7.1. DECISION-BOUNDARY DIFFERENCE ANALYSIS VIA REPRESENTATION-SPACE VISUALIZATION

We first extract all harmful samples detected by the method (2500 in this setup), and randomly sample from the remaining training samples to supplement the set to 10,000 samples, thus forming a subset. This subset is enforced to include all harmful samples, with the randomness of the random sampling controlled by $seed = 0$. Subsequently, this subset is input into the anchored model (the parameters of which are $\theta$) to extract feature representations. These features are first reduced to $d = 50$ dimensions via PCA, followed by t-SNE mapping, to obtain two-dimensional visualization results. To reduce visualization perturbations caused by random data augmentation, we switch the training set transformations to test/evaluation transformations as much as possible in our implementation.

In Section 4.2, we have observed that harmful samples are not concentrated in the "inter-class transition regions" of model representation space; instead, they are scattered within and around the clusters of various classes. This indicates that the decision boundaries of human annotators and those of the model are not entirely consistent, and further demonstrates that "human mislabeling" does not necessarily correspond to "model inseparability". Furthermore, Fig. 6(a)-(c) visualize only the harmful samples: When colored by ground-truth labels, harmful samples still exhibit good separability in the model representation space; whereas when colored by noisy labels, they are clearly disorganized. After recoloring with our calibrated labels, their clustering structure can be restored to a pattern approximately consistent with that of ground-truth labels, thus verifying the effectiveness of the proposed method in performing label calibration on such "correctable mislabeling" samples.

### A.7.2. PROTOTYPE SIMILARITY HEATMAP ANALYSIS

To characterize the inter-class similarity structure from the model perspective, we extract the same feature representations on the trusted dataset $\widetilde{D}$ (with the anchor model parameters being $\tilde{\theta}$). Subsequently, samples features in $\widetilde{D}$ are grouped by class according to their labels; for each class, we sample at most 2000 samples to calculate the feature mean as the class prototype, and finally plot a cosine similarity heatmap between prototypes.

Synthesizing Fig. 5 and Fig. 7 reveals that: confusable class pairs to humans (e.g., Deer/Horse) are not necessarily equally similar in the model representation space; conversely, relatively distinguishable class pairs to humans (e.g., Cat/Dog) exhibit high similarity in model representations. This indicates that there exist differences in the discriminative cues relied on by the model and human annotators, and thus provides intuitive evidence for the notion that decision boundaries of human annotators and those of the model are not consistent.

### A.8. The EMVC Method Based on Purified Dataset

The core steps of the proposed data purification framework include dataset purification, population initialization, fitness evaluation, selection, crossover, and mutation.

***Dataset purification:*** First, in accordance with the gradient and feature dual-perception framework described in Section 3.1,

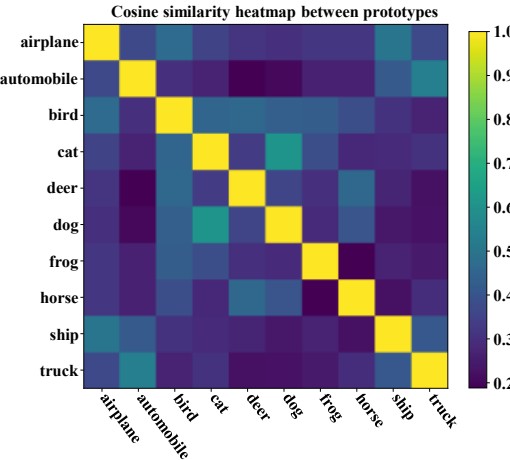

*Figure 7.* A similarity heatmap of class-wise feature prototypes obtained by feeding the trusted dataset $\widetilde{D}$ into the anchor model (with parameters $\tilde{\theta}$).

the given original dataset $D$ is processed to obtain the final purified dataset $D^\star$, which serves as the training data for the subsequent evolutionary process.

*Population initialization:* After obtaining the purified dataset $D^\star$, we construct the initial population on the search space composed of views and basic fusion operators. Specifically, we randomly generate $k$ individuals to form the population $P$. Each individual corresponds to a binary tree: leaf nodes denote views, and internal nodes denote basic fusion operators. The basic fusion operators adopted in this paper are from literature (Liang et al., 2021), including addition, multiplication, concatenation, maximize, and average. Due to the structural property of binary trees, if an individual contains $\mathcal{V}$ views, it must include $\mathcal{V} - 1$ fusion operators to ensure that all views contained in the individual are ultimately aggregated into a global representation.

*Fitness evaluation:* Given an individual in the population, it is first decoded into the corresponding multi-view model (MVM). In this MVM, each view first passes through a view encoder, mapping the original features to a unified fusion dimension $K$ to facilitate subsequent fusion. Subsequently, the feature vectors of each view are gradually combined through the corresponding fusion operators in accordance with the predefined fusion order in the structure of the individual, ultimately obtaining the global representation of the individual. This global representation then passes through a fully connected layer with an output dimension equal to the number of classes $C$ to generate a logits vector. After softmax, the class probability distribution is obtained, and the cross-entropy loss is calculated by comparing it with the sample labels. The finally obtained test accuracy is used as the fitness value of the individual. The above process corresponds to the most classic fitness evaluation paradigm in EMVC. While its subsequent derivative methods (e.g., EFB-EMVC (Liang et al., 2025c)) may exhibit differences in terms of loss function forms and training details, they generally still adhere to a unified evaluation framework consisting of "individual decoding → view dimensionality mapping → stepwise fusion according to the structure → obtaining the global representation → outputting predictions and calculating performance metrics accordingly".

Notably, $D^\star$ is cleaner than $D$. Therefore, the fitness values obtained by testing individuals trained on $D^\star$ can more accurately reflect the true performance of individuals, thereby alleviating FEB caused by label noise to a certain extent.

*Selection:* Binary tournament selection is adopted as the individual selection mechanism (Liang et al., 2021). The specific approach is as follows: randomly select two individuals from the candidate set each time, compare their fitness values, regard the one with higher fitness as the "winner", and allow it to participate in subsequent operations accordingly. In the crossover operation, the binary tournament selection is performed twice to select two individuals as parent individuals; in the mutation operation, it is performed once to select one individual as the parent individual; when generating the next-generation population, merge the current population with the newly generated offspring population, perform the binary tournament selection $k$ times, and form the next-generation population with the selected individuals.

*Crossover and mutation:* During the evolutionary process, we perform structural updates on individuals in the population with a crossover probability $r_1$ and a mutation probability $r_2$, respectively, where $r_1 > r_2$ usually holds, conforming to the classic evolutionary algorithm setting of prioritizing crossover over mutation. In the crossover operation, we take the two

selected binary trees as parents and randomly select a non-root node from each tree as the crossover point. Subsequently, the node and its corresponding subtree are cut off from the original tree, swapped with the corresponding subtree in the other tree, and the swapped subtree is reconnected to the parent node of the original crossover point in the other tree. Thus, two new binary tree structures, i.e., two offspring individuals, are obtained. In the mutation operation, we first randomly sample a node from the selected binary tree structure: if the node is a branch node, a new operator is randomly selected from the predefined set of fusion operators to replace the original one; if the node is a leaf node, a new view is randomly chosen from the view set to replace the current view.

Algorithm 1 outlines the proposed data purification framework.

---

**Algorithm 1** Data Purification Framework

---

1: **Input:** Original training dataset $D = (X, Y)$; anchor model; test dataset $\hat{D} = (\hat{X}, \hat{Y})$; set of basic fusion operators $F$; number of iterations $T$.
2: **Output:** Purified training dataset $D^\star$; satisfactory MVM and its corresponding test accuracy.
3: **Data purification:** Apply the gradient and feature dual-perception data purification framework to $D$, and obtain the purified training dataset $D^\star$.
4: **Population initialization:** Generate an initial population $P_0$;
5: **Fitness evaluation:** Train each individual in $P_0$ on $D^\star$, and compute its fitness values on $\hat{D}$.
6: **for** $t = 1$ **to** $T$ **do**
7:     Generate offspring $Q_t$ using the crossover operator;
8:     Conduct mutation on each individuals in $Q_t$;
9:     Obtain the fitness values of all individuals in $Q_t$;
10:     Select next generation population $P_{t+1}$ from $Q_t \cup P_t$ using a selection operator;
11: **end for**
12: **Return** $D^\star$, satisfactory MVM and its corresponding test accuracy.

---

### A.9. Verifying the Existence of FEB from a Dynamic Evolution Perspective

Although we have demonstrated from the perspective of static results in Section 4.4.1 that label noise can induce fitness evaluation bias (FEB), this conclusion is still not intuitive enough to reveal how label noise gradually misleads the evolutionary direction during evolutionary iterations. To address this, this section further proceeds from the dynamic distributions of views and fusion operators in the evolutionary process, statistically analyzes the changes in the proportions of different views and fusion operators in the population with iterations, and intuitively verifies the existence of FEB with process-based evidence.

We take the evolutionary process of EDF on the first fold of data from the MVoxCeleb dataset as an example, where the number of population iteration rounds is set to 20. Fig. 8 illustrates the dynamic changes in the proportions of different views and fusion operators within the population as EDF evolves over iterations.

**1) Fusion operators**: Dominant operators differ under *Original* and *Purified* conditions. Under the *Original* condition (Fig. 8a), the population is ultimately dominated by the Avg operator (accounting for 76.9%), with the Add operator ranking second (20.0%), indicating that the evolutionary search converges to a region centered on Avg. Relatively, under the *Purified* condition (Fig. 8b), the Add operator becomes dominant (accounting for 71.4%), with the Avg operator ranking second (23.8%), and the evolutionary search converges to a region centered on Add.

**2) View selection**: Converged view combination regions differ under *Original* and *Purified* conditions. Under the *Original* condition (Fig. 8c), the evolutionary process ultimately converges to regions containing views 1, 2, 3, 4; while under the *Purified* condition (Fig. 8d), the evolutionary process converges to regions containing views 1, 2, 4, showing distinctly different convergence results from those under the *Original* condition.

From the comprehensive analysis of the aforementioned dynamic distributions, it can be observed that under the two data conditions of *Original* and *Purified*, the convergence directions of evolutionary search and the solution space regions where the search finally stabilizes exhibit significant differences. This phenomenon intuitively indicates that label noise can affect evolutionary trajectories, thus giving rise to the FEB problem.

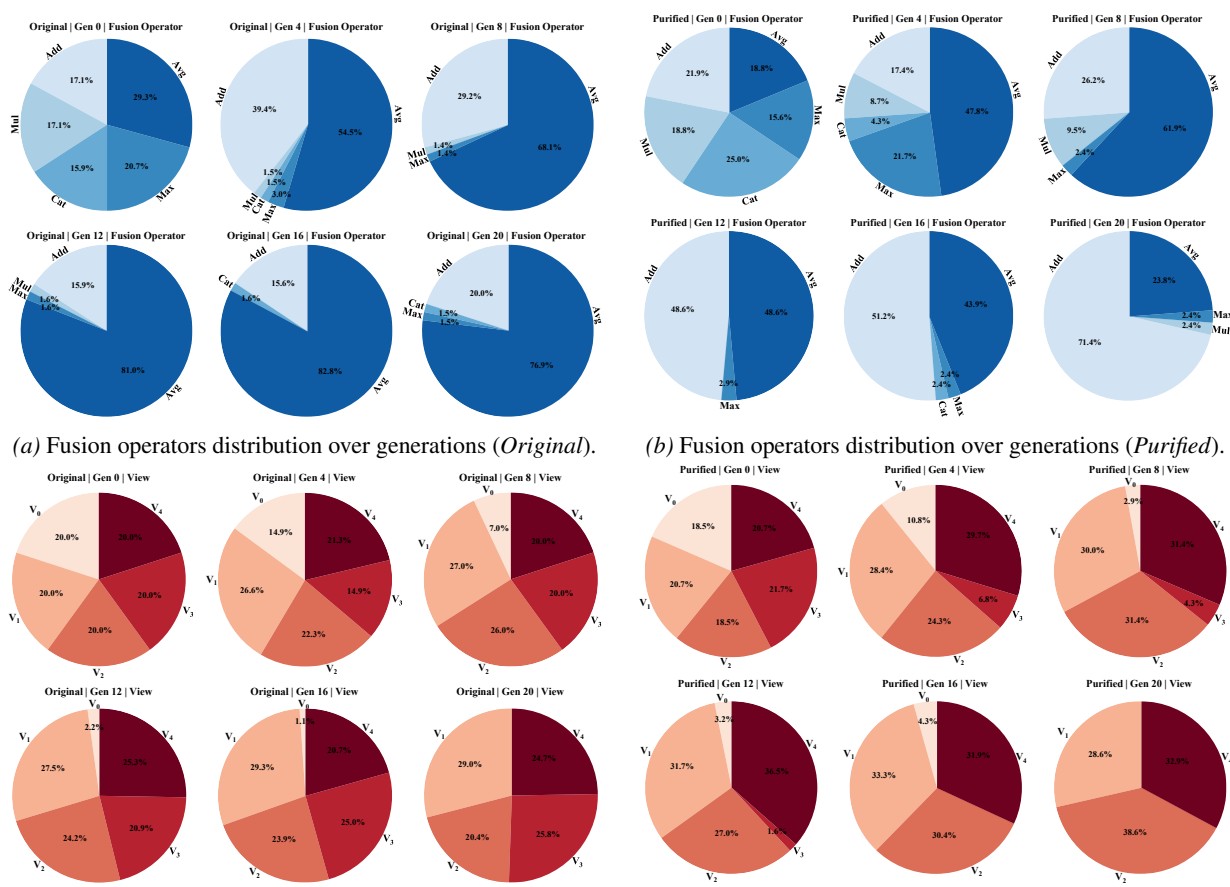

*(a)* Fusion operators distribution over generations (*Original*).

*(b)* Fusion operators distribution over generations (*Purified*).

*(c)* Views distribution over generations (*Original*).

*(d)* Views distribution over generations (*Purified*).

*Figure 8.* Dynamic evolution of the proportions of views and fusion operators under the *Original/Purified* settings.

