# OpenReview forum: "GFD-EMVC: Evolutionary Multi-View Classification with Label Noise via Gradient and Feature Dual-Perception"
_ICML.cc/2026/Conference — ICML 2026 spotlight_

### Official Review · Reviewer_peCf · 2026-02-23

**Soundness:** 4
**Presentation:** 4
**Significance:** 3
**Originality:** 3
**Overall Recommendation:** 5
**Confidence:** 5

**Summary:**

This paper proposes a simple yet effective data purification framework that focuses on the key issue of fitness evaluation bias (FEB) induced by label noise in the scenario of evolutionary multi-view classification (EMVC). To address this challenge, the authors integrate two mechanisms, namely outlier detection in the gradient space and prototype calibration in the feature space, to perform purification on training data. Experimental results demonstrate that the method yields consistent performance improvements across different noise intensity settings on real-world label noise datasets; it also significantly mitigates the evaluation distortion caused by FEB under noisy conditions, thereby reducing the negative impact of label noise on the evolutionary search process.

**Compliance With Llm Reviewing Policy:**

Affirmed.

**Final Justification:**

All my concerns have been addressed.

**Key Questions For Authors:**

See the above weaknesses.

**Limitations:**

yes

**Strengths And Weaknesses:**

***Strengths***

(1) The proposed data purification pipeline is simple and intuitive, with a clear implementation path, which lowers the barrier to implementation and reproduction. It can also be readily integrated as a preprocessing module into existing EMVC pipelines, offering strong engineering compatibility.

(2) The implementation details are described thoroughly, including the hyperparameter settings for both baselines and the proposed method, as well as key implementation choices such as individual encoding in the FEB experiments.

(3) The experimental argumentation is fairly rigorous. In prior work, once changes in individual ranking and overall performance gains were observed, these improvements were directly attributed to alleviated FEB. In contrast, this paper goes further in Section 4.4.2 by considering and ruling out the alternative explanation that performance gains stem merely from cleaner data while the evolutionary search does not actually converge to a better solution. The additional controlled comparison in Fig. 4(a) further strengthens the claim that the proposed method mitigates FEB and guides evolutionary search toward better optima.

***Weaknesses***

(1) In the noisy-label experiments on multi-view datasets, the paper mainly considers random label-flip noise. It is unclear why other noise types are not explored, such as class-dependent noise (e.g., flipping labels to semantically similar classes). I would appreciate it if the authors could clarify the rationale for this choice.

(2) Some descriptions are not sufficiently clear. In Section 3.2, the authors state that “each view is mapped to a unified dimension K via an encoder,” but it is unclear what exactly the “encoder” refers to. Is it a standard deep backbone (e.g., ResNet-18), or do different views use different architectures? I suggest clarifying this point and providing the corresponding implementation details.

---

> ### Author Rebuttal · Authors · 2026-03-27
>
> Thank you very much for your recognition of our work. We especially appreciate your positive comments on our focus on the core issue of FEB and on the rigor of our experimental validation.
>
> **W1: In the noisy-label experiments on multi-view datasets, the paper mainly considers random label-flip noise. It is unclear why other noise types are not explored, such as class-dependent noise (e.g., flipping labels to semantically similar classes). I would appreciate it if the authors could clarify the rationale for this choice.**
>
> A1: In this work, we adopt random label-flip noise for two main reasons.
>
> First, random label-flip noise is one of the most commonly used and standard experimental settings in label noise learning, and has been widely adopted in the modeling and experimental evaluation of related methods.
>
> Second, our method is based on the observation that the decision boundaries of human annotators are not entirely consistent with those of the model. Therefore, this work mainly focuses on mislabeled samples that are confusing to human annotators but still separable in the model representation space. Regarding class-dependent noise, on the one hand, our experiments are conducted on pre-extracted multi-view features rather than raw modal inputs, making it difficult to accurately characterize the true semantic similarity among classes and to construct representative class-dependent noise accordingly. On the other hand, if the noise transition directions are artificially designed according to class similarity in the current model representation space, the resulting noisy samples would typically already be inherently difficult to distinguish in the model representation space. These situations are not fully aligned with the problem setting considered in this work. Based on the above considerations, we adopt random label-flip noise as the main experimental setting.
>
> In future work, we plan to conduct human re-annotation on multimodal datasets with accessible raw samples and to construct noisy multi-view benchmark datasets that are closer to real-world scenarios, so as to further validate the effectiveness of our method.
>
> **W2: Some descriptions are not sufficiently clear. In Section 3.2, the authors state that “each view is mapped to a unified dimension K via an encoder,” but it is unclear what exactly the “encoder” refers to. Is it a standard deep backbone (e.g., ResNet-18), or do different views use different architectures? I suggest clarifying this point and providing the corresponding implementation details.**
>
> A2: In our work, the “encoder” does not refer to an existing DNN backbone such as ResNet-18, but rather to a lightweight feature mapping module; moreover, all views adopt the same network structure. This design is motivated by the fact that our model does not learn directly from raw modal data, but instead performs subsequent unified mapping and fusion based on these pre-extracted features.
>
> Specifically, for the input of each view, we first apply Batch Normalization to normalize the features so as to improve the stability of the training process. Then, Dropout is employed to reduce the risk of overfitting. After that, the input is fed into a fully connected layer with ReLU activation and mapped into a unified dimension $K$. This dimension corresponds to the fused-view dimension described in the paper, and its purpose is to project the representations of different views into a consistent feature space for subsequent view fusion.
>
> After multi-view fusion, the fused features are passed through Batch Normalization again, followed by two operations, namely sign-sqrt and L2 normalization. These operations help improve the stability of the subsequent classification learning. Finally, the normalized fused features are mapped into the class space through a fully connected layer, transformed into a probability vector via softmax, and then used to compute the cross-entropy loss with the labels, after which the model parameters are updated through backpropagation.
>
> Thank you for pointing out the issues regarding the choice of label noise types and the specific meaning of the encoder. We will add the relevant clarifications in the revised version to make the paper more complete and clear.

---

> > ### Author Rebuttal · Reviewer_peCf · 2026-04-01
> >
> > All my concerns have been addressed.

---

### Official Review · Reviewer_o8dz · 2026-03-09

**Soundness:** 3
**Presentation:** 3
**Significance:** 3
**Originality:** 3
**Overall Recommendation:** 5
**Confidence:** 5

**Summary:**

In this paper, the authors propose a "detect-then-calibrate" data purification framework. More specifically, the authors first perform outlier analysis in the gradient space, treating outliers as noisy samples; they then conduct prototype calibration in the feature space, utilizing feature prototypes of noise-free samples to correct the labels of noisy samples. The authors carry out comprehensive experiments on real-world label noise datasets as well as multi-view datasets with artificially injected label noise. Experimental results demonstrate that the proposed method can not only effectively purify data but also alleviate the search misguidance caused by fitness evaluation bias (FEB) in the scenario of evolutionary multi-view classification (EMVC).

**Compliance With Llm Reviewing Policy:**

Affirmed.

**Final Justification:**

This paper proposed a "detect-then-calibrate" data purification framework, which is novel and interesting. The rebuttal addressed my concerns and thus I keep the positive score.

**Key Questions For Authors:**

The authors have conducted sensitivity analysis. However, the clipping ratio is often difficult to determine in advance for practical applications. It is recommended to further verify its stable range across a wider parameter scope and more datasets.

**Limitations:**

yes

**Strengths And Weaknesses:**

Strengths
1.The paper is clearly and smoothly written, with a well-organized structure. The sections connect naturally, helping readers grasp the core ideas and the overall narrative.
2.The figures are clean and easy to follow, providing an intuitive illustration of the key challenges that existing methods face in noisy settings, and clearly outlining the overall pipeline and main components of the proposed approach.
3.The experimental design is fairly comprehensive. The authors provide corresponding empirical validation and support for most of the main conclusions, making the argumentation more convincing.
Weaknesses
1.Although the experimental evaluation is generally thorough, one potential limitation remains: the paper does not yet validate the method on real-world multi-view datasets with label noise. The authors also note in Appendix A.2 that such public datasets are currently lacking, so this issue is largely due to data availability rather than an oversight in experimental design. If the authors could construct or curate relevant benchmarks in the future, or add experiments on multi-view data with noise patterns closer to real-world settings, it would further strengthen the completeness and persuasiveness of the work.
2.The label correction in the second stage leverages class prototypes from the trusted set and cosine similarity for implementation, presenting a highly straightforward rationale. A potential issue, however, arises as follows: if a certain proportion of noise still remains in the trusted set, the prototypes will be contaminated, and such calibration runs the risk of further solidifying the existing errors. Though this is not a fatal flaw, it raises concerns that the stability of the method may be compromised in the face of some complex data distributions.

---

> ### Author Rebuttal · Authors · 2026-03-26
>
> We sincerely thank the reviewer for the positive evaluation of the paper’s writing, figure presentation, and experimental design.
>
> **W1: Although the experimental evaluation is generally thorough...**
>
> A1: Thank you for your suggestion. We agree that validation on real-world multi-view datasets with label noise would further strengthen the completeness and persuasiveness of this work. However, due to the current lack of publicly available real-world multi-view datasets with label noise, and because most existing multi-view benchmarks are released in the form of pre-extracted features, which makes it difficult to analyze real noise patterns or construct noise that more closely resembles human annotation behavior, we adopted the commonly used random flip strategy in noisy-label learning to generate noisy labels and validated the effectiveness of our method under this standard setting. In the future, we plan to conduct human re-annotation on multimodal datasets with accessible raw samples and construct noisy multi-view benchmarks that are closer to real-world scenarios, in order to further verify the performance of our method.
>
> **W2: The label correction in the second stage leverages class prototypes...**
>
> A2: In response to your concern, we have added an ablation study on the anchor model. In the multi-view experiments, the label noise rate is 40%, while the clipping ratio is only 20%, which means that the trusted set after detection still contains some noisy samples. Even so, the results in Table 1 show that Anchor model$^{+}$ consistently outperforms both the original Anchor model and Anchor model$^{-}$ across all multi-view datasets, indicating that prototype contamination does not undermine the purification effectiveness or stability of our method.
>
> Table 1: Ablation experiments of anchor model structure(%). Method: Training on the original dataset; Method$^{-}$: Training after removing detected noisy samples; Method$^{+}$: Training on the purified dataset.
>
> |Method|MVoxCeleb|YoutubeFace|Reuters5|Reuters3|NUS|
> |-|-|-|-|-|-|
> |Anchor model|73.28|78.80|74.96|80.01|68.45|
> |Anchor model$^{-}$|76.14|78.04|76.28|81.05|68.60|
> |Anchor model$^{+}$|79.18|80.58|77.09|81.57|69.69|
>
> **Q1: The authors have conducted sensitivity analysis...**
>
> A1: Following the reviewer’s suggestion, we supplemented the ablation experiments with a wider range of $\kappa$% values and more datasets. Table 2 shows that, for low-noise datasets, the best performance usually appears near the true noise rate; for high-noise datasets, the best results within our current search range are typically achieved at relatively large $\kappa$ % values.
>
> Table 2: Test accuracy (%) under different clipping ratios $\kappa$% on nine datasets.
>
> |$\kappa$ %| CIFAR-10N(Agree) | CIFAR-10N(Rand1) | CIFAR-10N(Worst) | CIFAR-100N(Noisy100) | MVoxCeleb | YoutubeFace | NUS | Reuters5 | Reuters3 |
> |---|---|---|---|---|---|---|---|---|---|
> |5.0%|92.16|90.42|84.20|61.21|73.88|78.64|68.64|75.47|80.21|
> |7.5%|92.17|90.78|84.14|61.60|74.73|79.58|68.47|75.47|80.52|
> |10.0%|**92.21**|91.13|84.12|61.80|75.74|79.51|69.01|76.04|80.76|
> |12.5%|92.15|91.20|84.66|**62.24**|76.64|80.13|69.39|76.43|80.94|
> |15.0%|91.72|91.37|84.85|61.30|77.55|80.33|69.28|76.44|81.18|
> |17.5%|91.75|91.35|**85.18**|61.74|78.29|**80.75**|69.56|77.08|81.23|
> |20.0%|91.70|**91.54**|85.10|62.01|**79.18**|80.58|**69.69**|**77.09**|**81.57**|
>
> In response to the reviewer’s valuable suggestions, we will incorporate the above discussion and additional ablation studies into the revised manuscript.

---

> > ### Author Rebuttal · Reviewer_o8dz · 2026-04-01
> >
> > The rebuttal addressed all my concerns.

---

### Official Review · Reviewer_HXK7 · 2026-03-12

**Soundness:** 4
**Presentation:** 4
**Significance:** 4
**Originality:** 3
**Overall Recommendation:** 5
**Confidence:** 5

**Summary:**

This paper focuses on an easily overlooked phenomenon in the EMVC scenario: once erroneous annotations are introduced into the training set, the performance metrics relied on by the evolutionary process become perturbed, which causes distortion in the ranking of individual performance and gives rise to fitness evaluation bias (FEB), ultimately misleading the search direction. To address this issue, the authors introduce a data processing pipeline prior to evolutionary search: potential problematic samples are first screened out by virtue of anomalous gradients generated during sample training, and then such samples are recalibrated instead of being simply discarded by leveraging the class prototypes formed by reliable samples in the representation space. Extensive experiments verify that the method proposed by the authors can yield consistent performance gains across different noise intensity levels and mitigate the impact of FEB on EMVC.

**Compliance With Llm Reviewing Policy:**

Affirmed.

**Final Justification:**

After comprehensive consideration, I assign this score.

**Key Questions For Authors:**

Although the ablation studies are relatively thorough, the anchor model is directly set to “all-view additive fusion.” This raises a natural question: how would the ablation results change if the anchor model used other fusion operators (e.g., concatenation, multiplicative fusion, etc.)? It is better to include additional ablations to investigate this.

**Limitations:**

yes

**Strengths And Weaknesses:**

Strengths
(1)The authors plan to release their source code, which will be helpful for others who wish to reproduce the results or build upon this work.

(2)Overall, the paper is well written. The supplementary material also provides sufficient details to complement parts that could not be fully elaborated in the main text (e.g., the specific workflow of EMVC), which is particularly helpful for readers who want a deeper understanding of EMVC.

(3)The experiments are fairly comprehensive. For example, the ablation studies are broad in scope, covering both the label-noise learning setting and the multi-view learning setting.

Weaknesses
(1)It requires readers to cross-reference the meanings of certain notations in the paper—such as set symbols, clipping ratios and population sizes—across different paragraphs. It is better to provide a unified notation glossary would significantly enhance the readability of the paper.

(2)The discussion of EMVC in the related work section is somewhat brief. It only mentions that existing studies focus on search efficiency and FEB caused by view imbalance, without providing a more detailed description, which may be less accessible to researchers outside this area.

(3)Although the ablation studies are relatively thorough, the anchor model is directly set to “all-view additive fusion.” This raises a natural question: how would the ablation results change if the anchor model used other fusion operators (e.g., concatenation, multiplicative fusion, etc.)? It is better to include additional ablations to investigate this.

---

> ### Author Rebuttal · Authors · 2026-03-25
>
> We sincerely thank the reviewer for the positive assessment and for recognizing our key insight that erroneous annotations can induce FEB in EMVC, as well as the effectiveness of our method.
>
> **W1: It requires readers to cross-reference...**
>
> A1: Following your suggestion, we have added Table 1 to summarize the main notations. Due to space limitations, the current table includes only the key symbols and their meanings, and we will provide a more comprehensive notation glossary in the revised version.
>
> Table 1: Notation glossary.
>
> |Notation|Meaning|
> |-|-|
> |$D$|Original training dataset|
> |$\mathcal{N}$|Noisy training set|
> |$\widetilde{D}$|Trusted training set|
> |$D^\star$|Purified training dataset|
> |$\hat{D}$|Test dataset|
> |$P$|Population|
> |$k$|Population size|
> |$K$|Dimension of the fused view|
> |$\mathcal{V}$|Number of views|
> |$C$|Number of classes|
> |$c$|The $c$-th class|
> |$\mathbf{z}_i$|Fused feature vector of the $i$-th sample|
> |$\boldsymbol{\mu}_c$|Prototype of the $c$-th class|
> |$\theta$|Optimal anchor model parameters trained on the original dataset|
> |$\tilde{\theta}$|Optimal anchor model parameters trained on the trusted set|
> |$\kappa$%|Clipping ratio|
>
> **W2: The discussion of EMVC in the related work section...**
>
> A2: The current version of the paper has already outlined the main research directions in EMVC, but due to space limitations, the description is relatively brief. Following your suggestion, we have provided a more detailed description to improve accessibility for researchers outside this area. EMVC leverages the adaptive mechanism of evolutionary algorithm (EA) to iteratively optimize and select high-performing multi-view model (MVM) from the population. Its pioneering work, EDF [1], was the first to obtain satisfactory MVM through evolutionary search in a space composed of views and basic fusion operators. Subsequent studies mainly focused on improving search efficiency and model credibility: DC-NAS [2] adopts a divide-and-conquer data strategy to accelerate the search process; CSG-NAS [3] reduces search complexity by constructing a core-structure search space; KS-NAS [4] further lowers computational cost by introducing a dynamic knowledge base; and TEF [5] uses EA to generate high-quality pseudo-views, thereby enhancing the credibility of MVM. Although existing methods have achieved substantial progress, none of them takes the FEB problem into account. To this end, EFB-EMVC [6] is the first to identify this issue and, starting from the view imbalance phenomenon, introduces evolutionary navigators to explicitly guide the training of each view branch, yielding remarkable performance improvements.
>
> [1] IEEE TEVC, 2021. [2] AAAI, 2024. [3] IJCAI, 2024. [4] IJCAI, 2025. [5] ICLR, 2025. [6] NeurIPS, 2025.
>
> **W3: Although the ablation studies are relatively thorough...**
>
> A3: Following your suggestion, we added ablation experiments on different anchor model structures across multiple multi-view datasets. The results in Table 2 show that most fusion operators achieve favorable purification performance across different datasets, indicating that our method is generally adaptable to different anchor model structures and therefore exhibits good generalizability. It is worth noting that Mul performs poorly on NUS (28.56/23.34/27.30), mainly because of its weak discriminative ability on the original noisy dataset, which makes it difficult to learn stable fused representations, thereby affecting noisy sample detection and subsequent correction, and ultimately limiting the purification effect.
>
> Table 2: Ablation experiments of anchor model structure(%). Method: Training on the original dataset; Method$^{-}$: Training after removing detected noisy samples; Method$^{+}$: Training on the purified dataset.
>
> |Method|MVoxCeleb|YoutubeFace|Reuters5|Reuters3|NUS|
> |-|-|-|-|-|-|
> |Add|73.28|78.80|74.96|80.01|68.45|
> |Add$^{-}$|76.14|78.04|76.28|81.05|68.60|
> |Add$^{+}$|79.18|80.58|77.09|81.57|69.69|
> |Mul|72.50|79.08|72.11|78.23|**28.56**|
> |Mul$^{-}$|75.17|77.00|74.81|80.71|**23.34**|
> |Mul$^{+}$|78.09|79.87|76.27|81.40|**27.30**|
> |Cat|66.98|78.80|75.17|80.45|66.92|
> |Cat$^{-}$|69.76|78.13|75.70|80.81|66.87|
> |Cat$^{+}$|72.66|80.42|76.27|81.79|68.32|
> |Max|64.74|77.62|74.90|79.72|67.04|
> |Max$^{-}$|67.61|77.68|76.32|81.04|67.13|
> |Max$^{+}$|70.61|79.59|77.09|81.86|68.98|
> |Avg|72.96|77.63|73.22|78.73|65.34|
> |Avg$^{-}$|75.92|78.59|74.72|79.86|64.68|
> |Avg$^{+}$|79.11|80.32|75.40|80.48|67.38|
>
> Thank you very much for your valuable suggestion. We will include this additional ablation study and the corresponding analysis in the revised version to make the experimental evaluation and discussion more complete.

---

> > ### Author Rebuttal · Reviewer_HXK7 · 2026-04-02
> >
> > Since my problem has been resolved, I will continue to maintain my positive rating.

---

### Official Review · Reviewer_zKuZ · 2026-03-12

**Soundness:** 3
**Presentation:** 3
**Significance:** 3
**Originality:** 3
**Overall Recommendation:** 4
**Confidence:** 3

**Summary:**

This paper reveals that the decision boundaries of human annotators and models are not entirely aligned-samples indistinguishable to humans still exhibit separability and calibrability under the model's feature representations. This finding overturns the practice of "directly discarding detected noisy samples", preventing the loss of valuable complementary information in multi-view data through sample discard. Building upon this insight, the paper proposes a simple yet efficient "detect-then-calibrate" data purification framework with dual perceptual capabilities for gradients and feature.

**Compliance With Llm Reviewing Policy:**

Affirmed.

**Final Justification:**

My concerns have been partially resolved, and I will maintain my original score.

**Key Questions For Authors:**

1. Label noise in real-world scenarios predominantly occurs on hard-to-classify samples near human decision boundaries and is often associated with multi-view complementary features (e.g., samples distinguishable in some views but ambiguous in others are more prone to mislabeling). How does the performance of the proposed method change on datasets that more closely resemble real-world scenarios?

2. The proposed method recalibrates labels for all detected outlier samples without implementing a confidence filtering mechanism. This process may introduce new labeling errors-does this further exacerbate FEB?

Please also refer to the Weaknesses.

**Limitations:**

Yes.

**Strengths And Weaknesses:**

**Strengths:**

1. In the EMVC scenario, this study reveals and analyzes how label noise induces FEB at the data level, enriching the research framework on FEB within the EMVC domain. Subsequently, inconsistencies between human and model decision boundaries were identified, leading to the design of a dual-perception data purification framework. Compared to traditional strategies, this approach more effectively leverages the valuable information within ambiguous samples.

2. Extensive experiments thoroughly validate the effectiveness of the proposed method. On real noisy labeled datasets CIFAR-10N and CIFAR-100N, it achieves optimal performance compared to mainstream label noise learning methods. The existence of the FEB phenomenon induced by label noise in multi-view scenarios is confirmed, and it is demonstrated that data purification can effectively mitigate FEB.

**Weaknesses:**

1. This paper employs a fixed-structure anchor model with "full-view additive fusion" to extract gradient and feature representations, but fails to adequately demonstrate the rationale for selecting the anchor model structure. This paper does not verify the adaptability of the anchor model structure to multi-view datasets with different complexities, and cannot determine whether this anchor model can effectively extract discriminative gradients and features from high-dimensional and heterogeneous multi-view data. In addition, this paper does not analyze the impact of the anchor model's performance on subsequent data purification. If the anchor model itself has poor classification performance on the original dataset, will its extracted gradients and features lead to biases in noise detection and label calibration? These aspects lack relevant ablation experiments for verification.

2. Although this paper explicitly states that the proposed method only applies to "correctable mislabeled samples" and struggles to reliably calibrate mislabeled samples simultaneously located at the model decision boundary, it fails to provide a clear definition and criteria for "mislabeled samples near the model decision boundary", making it impossible to distinguish this type of sample from "correctable mislabeled samples". This paper also fails to analyze the proportion of this type of sample in real-world data and its impact on EMVC performance. If this type of sample accounts for a high proportion, the actual effectiveness of the proposed method will significantly decrease. In addition, this paper does not propose any supplementary solutions, merely defining the scope of application. This results in insufficient completeness of the proposed method, making it unable to address complex real-world label noise scenarios.


3. In this paper, $\kappa$% is fixed at 5% for label noise learning and 20% for multi-view learning, and the sensitivity of $\kappa$% is analyzed only under the Aggregate noise setting of CIFAR-10N. This paper lacks an adaptive $\kappa$% design. In real-world scenarios, the true noise rate of the dataset is often unknown, and a fixed ratio is prone to undercalibration or overcalibration. This paper only suggests using 5% as an initial value without providing specific tuning methods or adaptive strategies. In addition, this paper does not validate the sensitivity of $\kappa$% on other noisy datasets (e.g., the Worst noise setting of CIFAR-10N and CIFAR-100N) and multi-view datasets, and therefore cannot determine the robustness of this parameter under different noise rate scenarios.

---

> ### Author Rebuttal · Authors · 2026-03-28
>
> Thank you for your positive comments.
>
> **W1: This paper...**
>
> A1: Following your suggestion, we added ablation experiments on different anchor model structures across multiple multi-view datasets. Table 1 shows that full-view additive fusion is more stable across datasets of different complexities, and we thus adopt it as the anchor model structure. In addition, the poor results of Mul on NUS (28.56/23.34/27.30) indicate that if the anchor model itself has poor classification performance on the original dataset, the extracted gradients and features may bias subsequent noisy-sample detection and label calibration, thus weakening data purification.
>
> Table 1: Ablation of anchor model structure(%). Method: Training on the original dataset; Method$^{-}$: Training after removing detected noisy samples; Method$^{+}$: Training on the purified dataset.
> |Method|MVoxCeleb|YoutubeFace|Reuters5(R5)|Reuters3(R3)|NUS|
> |-|-|-|-|-|-|
> |Add|73.28|78.80|74.96|80.01|68.45|
> |Add$^{-}$|76.14|78.04|76.28|81.05|68.60|
> |Add$^{+}$|79.18|80.58|77.09|81.57|69.69|
> |Mul|72.50|79.08|72.11|78.23|**28.56**|
> |Mul$^{-}$|75.17|77.00|74.81|80.71|**23.34**|
> |Mul$^{+}$|78.09|79.87|76.27|81.40|**27.30**|
> |Cat|66.98|78.80|75.17|80.45|66.92|
> |Cat$^{-}$|69.76|78.13|75.70|80.81|66.87|
> |Cat$^{+}$|72.66|80.42|76.27|81.79|68.32|
> |Max|64.74|77.62|74.90|79.72|67.04|
> |Max$^{-}$|67.61|77.68|76.32|81.04|67.13|
> |Max$^{+}$|70.61|79.59|77.09|81.86|68.98|
> |Avg|72.96|77.63|73.22|78.73|65.34|
> |Avg$^{-}$|75.92|78.59|74.72|79.86|64.68|
> |Avg$^{+}$|79.11|80.32|75.40|80.48|67.38|
>
> **W2: Although this... & Q2: The proposed...**
>
> A2: We define mislabeled samples near the model decision boundary as mislabeled samples located around inter-cluster boundaries in the model representation space, which cannot be reliably distinguished by the model, and define correctable mislabeled samples as those located within clusters and exhibiting stronger separability. Based on the observation that human and model decision boundaries are not fully aligned, our method is designed to correct the latter type of samples and achieves strong performance. We have also added Table 2 to report the proportion of truly mislabeled samples among the detected outliers, as well as the correction accuracy and error rate. Table 2 results and the overall performance of the proposed method indicate that the former type of samples does not dominate in the current datasets; therefore, its impact on the EMVC performance is limited. Meanwhile, even without introducing a confidence filtering mechanism, the proposed method does not exacerbate FEB. In future work, we will investigate correction strategies for the former type of samples and confidence filtering mechanisms.
>
> Table 2: Detection and correction performance(%).
> ||CIFAR-10N(Agree)|CIFAR-10N(Rand1)|CIFAR-10N(Worst)|CIFAR-100N(Noisy100)|MVoxCeleb|YoutubeFace|R5|R3|NUS|
> |-|-|-|-|-|-|-|-|-|-|
> |True mislabel ratio|84.84|96.56|90.28|87.76|95.91|89.58|91.17|93.85|89.39|
> |Correction accuracy|81.49|93.88|70.81|63.85|81.98|79.42|77.37|83.11|70.37|
> |Correction error rate|18.51|6.12|29.19|36.15|18.02|20.58|22.63|16.89|29.63|
>
> **W3: In this...**
>
> A3: Following your suggestion, we added ablation studies of $\kappa$% over a wider range and on more datasets, using 5% as the starting point and tuning $\kappa$% with a step size of 2.5%. Table 3 shows, for low-noise datasets, the best performance usually appears near the true noise rate, while for high-noise datasets, the best results correspond to relatively large $\kappa$% values, indicating that this tuning strategy does not lead to severe under- or over-calibration issues. In addition, $\kappa$% remains relatively stable across different noise scenarios, indicating a certain degree of robustness. A more adaptive strategy will be studied in future work.
>
> Table 3: Accuracy(%) under different clipping ratios.
> |$\kappa$%|CIFAR-10N(Agree)|CIFAR-10N(Rand1)|CIFAR-10N(Worst)|CIFAR-100N(Noisy100)|MVoxCeleb|YoutubeFace|NUS|R5|R3|
> |-|-|-|-|-|-|-|-|-|-|
> |5.0%|92.16|90.42|84.20|61.21|73.88|78.64|68.64|75.47|80.21|
> |7.5%|92.17|90.78|84.14|61.60|74.73|79.58|68.47|75.47|80.52|
> |10.0%|**92.21**|91.13|84.12|61.80|75.74|79.51|69.01|76.04|80.76|
> |12.5%|92.15|91.20|84.66|**62.24**|76.64|80.13|69.39|76.43|80.94|
> |15.0%|91.72|91.37|84.85|61.30|77.55|80.33|69.28|76.44|81.18|
> |17.5%|91.75|91.35|**85.18**|61.74|78.29|**80.75**|69.56|77.08|81.23|
> |20.0%|91.70|**91.54**|85.10|62.01|**79.18**|80.58|**69.69**|**77.09**|**81.57**|
>
> **Q1: Label noise...**
>
> A1: Results on existing multi-view datasets show that the anchor model can effectively fuse information from different views and achieve good correction performance. Therefore, on datasets that more closely resemble real-world scenarios, the proposed method is expected to leverage informative views to guide the learning of ambiguous views, thereby maintaining good correction performance.
>
> Thank you for your constructive suggestions. We will add relevant experiments and analyses in the revision.

---

> > ### Author Rebuttal · Reviewer_zKuZ · 2026-04-04
> >
> > The authors' rebuttal partially addressed my concerns, and I will maintain my rating.

---

### Decision · Program_Chairs · 2026-04-30

**Decision:**

Accept (spotlight)

**Comment:**

This submission was viewed favorably overall. The reviewers agreed that the paper addresses a meaningful and underexplored problem, fitness evaluation bias induced by label noise in evolutionary multi-view classification. The proposed detect-then-calibrate framework is technically sensible, simple to implement, and supported by broadly positive empirical results. The strongest points of consensus were the relevance of the problem, the practical appeal of combining gradient-space detection with feature-space calibration, the generally clear presentation, and a fairly comprehensive experimental study. The main concerns were about scope and completeness rather than core soundness: limited validation on real noisy multi-view data, reliance on random-flip noise in the multi-view setting, sensitivity to anchor-model design and clipping ratio, and some ambiguity around which mislabeled samples are actually correctable.